# An analgesic pathway from parvocellular oxytocin neurons to the periaqueductal gray in rats

Mai Iwasaki[1,12], Arthur Lefevre [1,2,11,12], Ferdinand Althammer[2,3,4,12], Etienne Clauss Creusot[1,12], Olga Łąpieś [1], Hugues Petitjean[1], Louis Hilfiger [1], Damien Kerspern[1], Meggane Melchior[1], Stephanie Küppers [2], Quirin Krabichler [2], Ryan Patwell [2], Alan Kania [2], Tim Gruber[5], Matthew K. Kirchner [3], Moritz Wimmer[4], Henning Fröhlich[4], Laura Dötsch[4], Jonas Schimmer[2], Sabine C. Herpertz[6], Beate Ditzen[7,8], Christian P. Schaaf[4,8], Kai Schönig[9], Dusan Bartsch[9], Anna Gugula [10], Aleksandra Trenk [10], Anna Blasiak [10], Javier E. Stern[3], Pascal Darbon[1], Valery Grinevich [2,3,13] ✉ & Alexandre Charlet [1,13] ✉

The hypothalamic neuropeptide oxytocin (OT) exerts prominent analgesic effects via central and peripheral action. However, the precise analgesic pathways recruited by OT are largely elusive. Here we discovered a subset of OT neurons whose projections preferentially terminate on OT receptor (OTR)-expressing neurons in the ventrolateral periaqueductal gray (vlPAG). Using a newly generated line of transgenic rats (OTR-IRES-Cre), we determined that most of the vlPAG OTR expressing cells targeted by OT projections are GABAergic. Ex vivo stimulation of parvocellular OT axons in the vlPAG induced local OT release, as measured with OT sensor GRAB. In vivo, optogenetically-evoked axonal OT release in the vlPAG of as well as chemogenetic activation of OTR vlPAG neurons resulted in a long-lasting increase of vlPAG neuronal activity. This lead to an indirect suppression of sensory neuron activity in the spinal cord and strong analgesia in both female and male rats. Altogether, we describe an OT-vlPAG-spinal cord circuit that is critical for analgesia in both inflammatory and neuropathic pain models.

The hypothalamic neuropeptide oxytocin (OT) modulates several key neurophysiological functions, including pain[1]. OT is produced in the hypothalamic supraoptic (SON) and paraventricular (PVN) nuclei by two major types of neurons: magnocellular (magnOT) and parvocellular (parvOT) neurons. MagnOT neurons of the SON and PVN are large cells that release OT into the bloodstream via axonal projections to the posterior pituitary. In contrast, parvOT neurons are smaller cells located exclusively in the PVN and project to the brainstem and spinal cord, but not the posterior pituitary[2]. It has been previously demonstrated that a small population of PVN parvOT neurons attenuates pain perception via two pathways: (1) through coordinated OT release into the bloodstream from magnOT neurons leading to the modulation of peripheral nociceptor activity in the dorsal root ganglion and skin and (2) by inhibiting sensory neurons in the spinal cord[3–5].

The periaqueductal gray (PAG) plays a pivotal role in descending analgesic pathways[6]. Indeed, physiological suppression of pain seems to be primarily modulated by a top-down system comprised of the PAG, rostral ventromedial medulla (RVM), and dorsal horn of the spinal cord (SC)[7]. For example, electrical stimulation of the PAG inhibits the firing rate of neurons in the dorsal horn of the spinal cord[8,9]. In

addition, both OT axons and OT receptors (OTR) have been reported in the PAG of mice[10,11], where the administration of exogenous OT enhances neuronal firing rates[12] and blocking of OTRs decreases pain threshold[13].

Altogether, these studies suggest that OT may promote analgesia through the OT-mediated activation of PAG neurons. It is therefore tempting to hypothesize two independent, yet complementary mechanisms of OT-mediated analgesia wherein OT attenuates nociceptive signals at the level of peripheral nociceptors and/or the spinal cord[3,14]. This would additionally act within the PAG to fine-tune additional descending pain-related pathways.

However, neither the cellular circuitry nor the analgesic effects of endogenous OT signaling in the PAG have been studied. To address this gap, we first generated a Cre knock-in rat line to label and manipulate OTR neurons in the PAG, wherein we observed both synaptic and non-synaptic contacts of OTergic axon terminals with somata and dendrites of OTR-positive PAG neurons. Next, we employed cell-type-specific viral vectors to identify a subpopulation of parvOT neurons projecting to the ventrolateral subregion of the PAG (vlPAG). We then used in vivo electrophysiology combined with optogenetics in the vlPAG to reveal that activation of OTR neurons leads to inhibition of sensory wide dynamic range (WDR) neurons in the spinal cord ($SC_{WDR}$) of anesthetized rats. Finally, we found that optogenetically-evoked OT release in the vlPAG produces analgesia and this effect was recapitulated by chemogenetic activation of vlPAG OTR neurons ($vlPAG_{OTR}$).

In this work, we identified an independent parvOT→$vlPAG_{OTR}$→$SC_{WDR}$ pathway that is distinct from the previously described direct parvOT→$SC_{WDR}$ pathway[3] and is capable of promoting analgesia in the context of both inflammatory and chronic neuropathic pain.

## Results

### vlPAG OTR-expressing neurons are GABAergic

To study OTR-expressing neurons in the vlPAG, we generated a transgenic line of rats with Cre recombinase expression controlled by the endogenous OTR gene locus (OTR-IRES-Cre line, see "Methods" for details) (Fig. 1a, Supplementary Fig. 1a−g). To label OTR neurons, we injected the PAG of OTR-IRES-Cre female rats with a rAAV carrying a Cre-dependent GFP expression cassette ($rAAV_{1/2}$-EF1α-DIO-GFP) ($n = 4$). We found a clustering of OTR neurons along the anteroposterior axis of the vlPAG (Supplementary Fig. 1d−g). To further assess the specificity of Cre localization in OTR-IRES-Cre rats, we performed RNA-Scope using probes against both OTR and Cre mRNAs ($n = 3$ rats) and found that 91.38% of Cre-positive cells were also positive for OTR mRNAs and 90.81% of OTR-positive cells also expressed Cre mRNAs (Fig. 1b, Supplementary Fig. 2a, b). We further validated the OTR-IRES-Cre rats by injecting $rAAV_{1/2}$-EF1α-DIO-GFP into the vlPAG combined with an antibody staining against Cre recombinase and found a 98.6% overlap between GFP and Cre signals (Supplementary Fig. 2c, d). In addition, we performed a western blot using the same Cre antibody and found the specific Cre band (35 kDa) only in OTR-IRES-Cre, but not wild-type rats (Supplementary Fig. 2e, f).

Ex vivo electrophysiology in acute brain slices of PAG showed that application of the selective OTR agonist, [Thr⁴Gly⁷]-oxytocin (TGOT), induced a significant increase in firing of GFP + OTR neurons which disappeared after washout (Baseline $0.573 \pm 0.295$ Hz vs TGOT $1.045 \pm 0.388$ Hz vs Wash $0.514 \pm 0.298$ Hz; $n = 11$; Fig. 1c−g). There was no response to TGOT in recorded GFP- neurons (Baseline $0.119 \pm 0.046$ Hz vs TGOT $0.108 \pm 0.049$ Hz vs Wash $0.122 \pm 0.064$ Hz, $n = 9$, Supplementary Fig. 1h, i). In addition, TGOT incubation induces a significant decrease of the first spike latency (FSL) only in GFP + OTR neurons ($n = 8/10$, Baseline $129.31 \pm 28.04$ ms vs TGOT $41.42 \pm 12.37$ ms, ***$p = 0.0041$ (paired two-tailed t test, $n = 10$, Fig. 1h−j) and has no global effect on the FSL of GFP-OTR neurons (Baseline $31.95 \pm 9.44$ ms

vs TGOT $37.08 \pm 14.89$ ms, $p = 0.788$ (paired two-tailed t test), $n = 7$, Supplementary Fig. 1j−m).

This result shows that OTR-IRES-Cre rats correctly express functional OTRs and Cre within the same cells and that the pharmacological activation of OTRs induces a significant change in the intrinsic excitability properties of GFP + OTR neurons.

We then quantified the number of vlPAG neurons expressing GFP and found that 396 out of 2135 (18.6%) cells were GFP-positive (Fig. 1k, l), indicating that about a fifth of all vlPAG cells express OTR. Histochemical analysis of vlPAG slices further revealed that the vast majority of GFP + cells stained positive for GAD-67, a marker of GABAergic neurons. This result indicates that virtually all of the vlPAG OTR neurons are GABAergic in nature (94.7%, $n = 174$ cells, Fig. 1m, n).

### A projection from PVN parvOT neurons to the vlPAG

To determine the origin of OT projections activating vlPAG OTR neurons, we injected a rAAV expressing mCherry under the control of the OT promotor ($rAAV_{1/2}$-OTp-mCherry[15], into either the PVN or SON in addition to a Cre-dependent rAAV expressing GFP ($rAAV_{1/2}$-EF1α-DIO-GFP) into the vlPAG of OTR-IRES-Cre female rats ($n = 4$; Fig. 2a). A schematic depiction of viral injections, expression time, treatment allocation and performed experiments can be found on Supplementary Fig. 3. We found mCherry+ axons in close proximity to GFP + OTR cells in the vlPAG after injection into the PVN (Fig. 2b−e). This indicates that axons from OT neurons within the vlPAG originate exclusively from the PVN. To further investigate a potential laterality of PVN→vlPAG fibers and to confirm injection sites, we injected additional rats ($n = 3$ male and $n = 3$ female) unilaterally into the PVN and SON with $rAAV_{1/2}$-OTp-GFP and $rAAV_{1/2}$-OTp-mCherry, respectively (Supplementary Fig. 4). Interestingly, we found that unilateral injections of the PVN resulted in robust OT fiber labeling in both hemispheres of the vlPAG, thus indicating that each PVN innervates both the left and right vlPAG. Here again, no red fibers were detected indicating that SON OT neurons do not project to the vlPAG.

We confirmed these results with retrograde tracing by injecting wild-type female rats ($n = 4$) with CAV2-CMV-Cre into the vlPAG and $rAAV_{1/2}$-OTp-DIO-GFP into the PVN (Fig. 2f). We found that only a few, relatively small ($n = 21$ cells, 10 to 20 μm diameter) OT neurons were labeled in the PVN (Fig. 2g−i). Furthermore, the retro-labeled cells were predominantly located at the latero-ventral edge of the PVN, although some GFP + neurons were found in the caudal medio-dorsal region of the PVN.

Because this discrete population resembled the morphology of parvOT neurons (e.g. small size and spindle-like shape), we injected wild type female rats ($n = 3$) with a marker of magnOT cells, Fluorogold (FG, Santa Cruz Biotechnology, Dallas, 15 mg/kg, i.p.), to specifically label magnOT, but not parvOT neurons[2]. In parallel, the same rats received an injection of green latex Retrobeads (Lumafluor Inc., Durham, NC, USA) into the vlPAG (Supplementary Fig. 5a) to retro-label only the OT neurons projecting to vlPAG. The histological analysis revealed that Fluorogold labeled PVN neurons did not contain the green puncta of Retrobeads in their cytoplasm (Supplementary Fig. 5b), indicating that the OT cell projections to the vlPAG represent parvOT neurons, but not magnOT neurons.

Next, we asked whether these ParvOT → $vlPAG_{OTR}$ neurons belonged to the same population of parvOT neurons we previously described[3] as projecting to the SON and spinal cord (SC). We first injected wild-type female rats ($n = 3$) with green Retrobeads into the vlPAG and red Retrobeads into the SC and found no OT + neurons in the PVN containing beads of both colors within the same cells (Supplementary Fig. 5c−f). We then analyzed whether ParvOT → $vlPAG_{OTR}$ neurons, identified in Fig. 2b, are projecting axons to the SC or SON

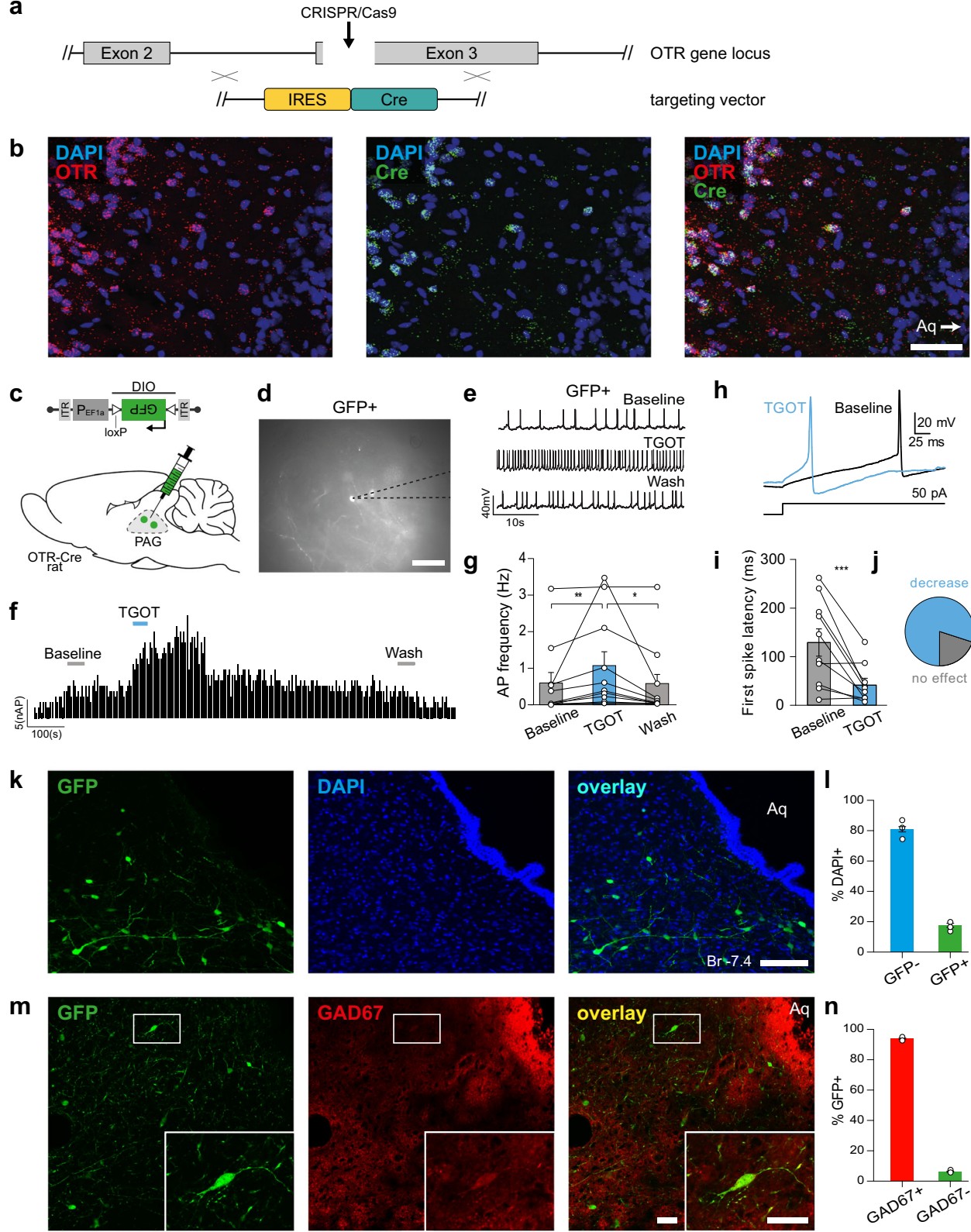

and found no detectable GFP + axons in the cervical, thoracic or lumbar segments of the SC, nor in the SON (Supplementary Fig. 5g, h). Finally, we injected another cohort of wild-type female rats ($n = 3$ per group) with CAV2-CMV-Cre into the SON (Supplementary Fig. 5i) or the SC (Supplementary Fig. 5l) and rAAV$_{1/2}$-OTp-DIO-GFP into the PVN. Here, we found that PVN neurons projecting to the SON also send axons to the SC, but not to the vlPAG (Supplementary Fig. 5j). Similarly,

PVN neurons projecting to the SC also send axons to the SON, but not to the vlPAG (Supplementary Fig. 5k, l). Altogether, our results indicate the existence of two distinct populations of parvOT neurons that project either to the vlPAG (present study) or to the SC[3]. A schematic distribution map highlighting the relative contributions of ParvOT neurons projecting to the vlPAG and spinal cord can be found on Supplementary Fig. 6.

**Fig. 1 | Generation of KI OTR-Cre rats and identification of vlPAG OTR neurons.**
**a, b** Generation of knock-in OTR-Cre rats; **a** Schema of the OTR gene locus and the insertion site of the IRES-Cre sequence. **b** RNA-scope in situ hybridization signal from Cre ($n_{cells}$ = 47 cells) and OTR ($n_{cells}$ = 42) probes. Scale bar = 200 μm.
**c–g** TGOT-induced increase of AP frequency of OTR neurons in the PAG of female rats. **c** Schema of rAAV$_{1/2}$-pEF1α-DIO-GFP injection in the PAG of OTR-Cre rats.
**d** Example image showing a GFP-positive cell during patch-clamp recordings. Scale bar = 20 μm. **e** Example traces from a GFP-positive cell under baseline, TGOT application, and wash out conditions. **f** Time course of GFP-positive cell activity (frequency distribution) upon TGOT application. **g** Quantification of TGOT effect on AP frequency of GFP-positive cells. Friedmann test, F = 14.97, $p < 0.0001$, $n_{cells}$ = 11, $n_{rats}$ = 4 followed by Dunn's multiple corrections: Baseline 0.573 ± 0.295 Hz vs TGOT 1.045 ± 0.388 Hz vs Wash 0.514 ± 0.298 Hz; **$p = 0.0029$, *$p = 0.0168$ (two-sided). **h–j** TGOT-induced change in first spike latency (FSL) of OTR neurons in the PAG. **h** Representative evoked currents in a GFP-positive

neuron in response to a square current step (50 pA) in baseline (black line) and after TGOT application (blue line). **i** FSL quantification for GFP-positive neurons showing the difference between baseline (129.31 ± 28.04 ms) and TGOT (41.42 ± 12.37 ms) conditions, ***$p = 0.0041$ (paired two-tailed t test, $n_{cells}$ = 10, $n_{rats}$ = 4). **j** Proportion of neurons after TGOT incubation with a decrease of the FSL superior to 10 ms ((blue) $n$ = 8/10) or with a variation of the FSL < 10 ms ((gray) $n$ = 2/10).
**k, l** Quantitative analysis of OTR cells in the PAG. **k** Images showing GFP (green) and DAPI (blue) staining of the vlPAG from OTR-Cre rats injected with rAAV$_{1/2}$-pEF1α-DIO-GFP virus. Scale bar = 100 μm. Aq = Aqueduct. **l** Bar plot showing the percentage of vlPAG cells expressing GFP ± SEM ($n_{cells}$ = 417, $n_{rats}$ = 3). **m, n** GAD67 staining of OTR cells in the vlPAG. **m** Image of a vlPAG brain slice stained for GFP (green) and GAD67 (red). Scale bar = 100 μm, inset scale bar = 20 μm. **n** Bar plot showing the percentage of GFP-positive neurons ± SEM ($n_{cells}$ = 338, $n_{rats}$ = 2) in the vlPAG stained and not stained by GAD67 antibody. Data are represented as mean ± SEM and as individual paired points. Source data are provided as a Source data file.

## PVN parvOT axons form somatic and dentritic contacts with vlPAG OTR neurons

Our next aim was to identify synaptic-like contacts between OT axons and OTR + neurons in the vlPAG. First, we counted the number of OT fibers in close proximity to OTR + neurons, relative to Bregma (Fig. 3a–c), and found a positive correlation between the two parameters ($R^2$ = 0.575, $p < 0.0001$, Fig. 3d), indicating that OT fibers specifically target OTR neurons in the vlPAG.

Next, we injected rAAV$_{1/2}$-pEF1α-DIO-GFP into the vlPAG of OTR-IRES-Cre female rats to label OTR neurons and then stained for OT, DAPI and the presynaptic marker synaptophysin (SYN, Fig. 3a, e, f). These sections were analyzed using Imaris software to quantify the OT innervation of GFP + (37%) and GFP- (4%) cells (Fig. 3g) as well as the percentage of synaptophysin-positive contacts between OT axons and GFP + somas (7%) or dendrites (56%) (Fig. 3h, Supplemental Fig. 7a-e). The latter indicated that the majority of OT axons predominantly form typical synaptic contacts on dendrites of PAG OTR + neurons similar to the OT-containing synapses previously demonstrated in the brainstem and SC[16]. When we quantified the proportion of SYN + OT fibers, we found that 80% of OTergic fibers at Bregma −6.5 mm, 90% of OTergic fibers at Bregma −7.5 mm and virtually 100% of OTergic fibers at Bregma −8.5 mm were positive for SYN +, thus essentially ruling out that these fibers further project to the spinal cord (Supplemental Fig. 7f).

Several reports have shown that OT is produced and released concomitantly with the conventional neurotransmitter, glutamate[15,17]. Therefore, we next tested whether OT-immunoreactive axons in the vlPAG also contained the glutamate transporter, vGluT2 (Fig. 3i, j). This analysis revealed that only 12.64% of OT fibers were also positive for vGluT2z (Fig. 3j). Importantly, we found synaptic-like contacts between GFP + dendrites and both vGluT2 + (Fig. 3k) and vGluT2- OT fibers (Fig. 3l). These findings suggest that a small percentage of direct PVN OT → vlPAG contacts are glutamatergic, although the precise role of glutamate in these putative synapses remains unclear.

## Evoked OT release in the vlPAG increases neuronal activity in vivo

Next we wanted to characterize the function of the PVN$_{OT}$ → vlPAG circuit in vivo by expressing channelrhodopsin2 (ChR2) fused with mCherry (rAAV$_{1/2}$-OTp-ChR2-mCherry; Fig. 4g) specifically in OT neurons of the PVN[3,15]. Firstly, we wanted to assess whether optogenetic stimulation of the OTergic axons arising from the PVN triggers release of OT within the vlPAG. To this end, we injected a modified channelrhodopsin2 (C1V1) fused with mCherry (rAAV$_{1/2}$-OTp-C1V1-mCherry), or mCherry alone as a control (rAAV$_{1/2}$-OTp-mCherry), in the PVN, and an OT biosensor named GRAB$_{OT}$[18] (rAAV$_{2/9}$-hSyn-OT1.0-sensor) in the vlPAG (Fig. 4a). The GRAB$_{OT}$ is a modified OTR with a fused cpGFP which becomes more fluorescent upon binding of OT to the receptor

(Fig. 4b). Next, vlPAG GRAB$_{OT}$ fluorescence was recorded in female rats ex vivo while stimulating the PVN OT fibers (30 s at 20 Hz, 30 ms pulse width) in presence or absence of atosiban, here used as an antagonist of GRAB$_{OT}$[18]. Overall, the stimulation of the PVN OTergic fibers within the vlPAG induced a significant increase of GRAB$_{OT}$ fluorescence (Fig. 4c, d), as shown by the increase of the maximum fluorescence value (mean ± SEM, control: 1.151 ± 0.352, C1V1: 7.987 ± 2.470, C1V1 + atosiban: 0.5843 ± 0.2279, $p = 0.0143$, $n$ = 16; Fig. 4e) and the increase of the area under the curve (mean ± SEM, control: 635.6 ± 150.2, C1V1: 2664 ± 632, C1V1 + atosiban: 409.7 ± 57.57, $p = 0.0045$, $n$ = 16; Fig. 4f).

In vivo PAG neuronal firing was then recorded in anesthetized female rats using silicone tetrodes coupled with a blue light (BL) that was used to stimulate PVN$_{OT}$ axons in the vlPAG (20 s at 30 Hz, 10 ms pulse width; Fig. 4g, Supplementary Fig. 8). Out of 82 recorded neurons, 21 showed an increase in firing rate (mean ± SEM; from 1.05 ± 0.39 to 17.65 ± 6.45 Hz, p = 0.0133; Fig. 4h–k). In contrast, two neurons, whose spontaneous activity prior to BL onset was high, showed a decreased firing rate within 300 s after the onset of BL (one cell from 25.83 to 6.95 Hz, another from 40.20 to 0.19 Hz; Fig. 4h, i). The remaining 59 neurons did not react to BL (Fig. 4h, i). We found the normalized mean activity of the excited neuronal population remained elevated for at least 300 s following the onset of BL (Fig. 4j). Notably, the time course of spike increase was diverse, as shown by the latency (1st quantile, median, 3rd quantile) for onset (1, 4, 40.25 s), peak activity (116.25, 155, 280.25 s) and offset (147.75, 296, 300) (Supplementary Fig. 8d). However, the total number of active neurons was maintained throughout the 300 s period following BL stimulation (Supplementary Fig. 8e). Therefore, we conclude that BL-evoked OT release in the vlPAG leads to an overall excitation of putative OTR + vlPAG neurons.

## Evoked OT release in vlPAG inhibits spinal cord WDR neurons activity in vivo

Next, we explored the downstream target of the PVN$_{OT}$ → vlPAG$_{OTR}$ circuit by performing in vivo BL stimulation of PVN$_{OT}$ axons in vlPAG (vlPAG-BL) while simultaneously recording sensory wide dynamic range (WDR) neuronal responses to electrical stimulation of the hind paw receptive field of female rats (Fig. 5a; Supplementary Fig. 9a). We focused on WDR neurons in the spinal cord (SC$_{WDR}$) because they are modulated by vlPAG inputs and have also been identified as an important cell population for integrating pain-related signals[19]. Indeed, peripheral sensory information converges from both fast (Aβ and Aδ type) and slow-conducting (C-type) primary afferent fibers, which are then integrated through WDR neurons in the deep laminae of the SC. Following repetitive electric stimulation to the hind paw WDR neuron receptive field, a short-term potentiation (wind-up; WU) occurs on the synapse made by C-type fibers onto WDR neurons that causes the spike rate of the cell to reach a plateau of maximal activity (Fig. 5b; time

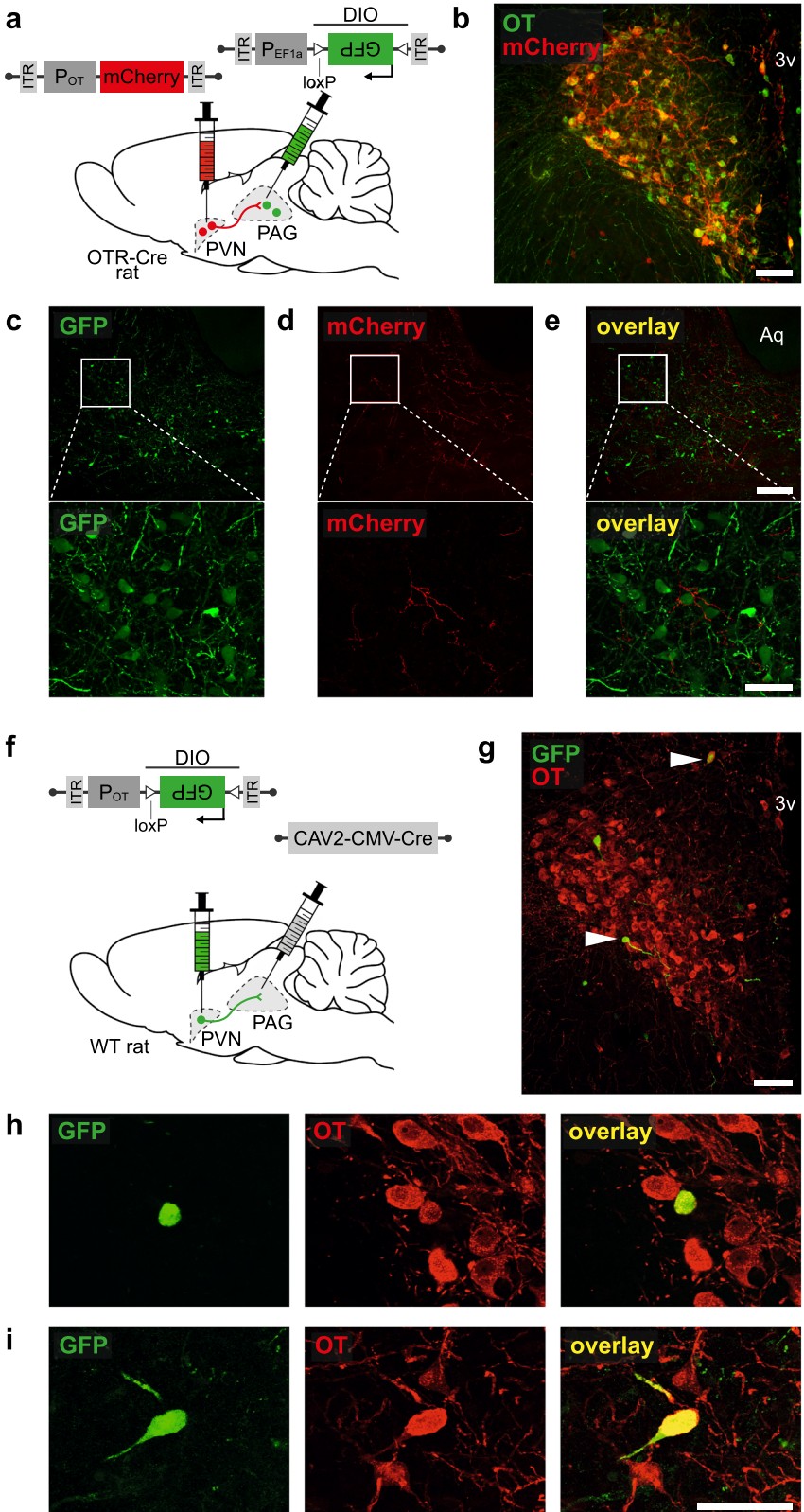

**Fig. 2 | PVN ParvOT neurons send axonal projections to vlPAG of female rats.**
**a–e** Anterograde tracing of projections from PVN OT-neurons to the vlPAG.
**a** Schema of viral injection showing injection of rAAV$_{1/2}$-pOT-mCherry in the PVN and rAAV$_{1/2}$-pEF1α-DIO-GFP in the PAG of OTR-Cre rats ($n = 4$ female rats). **b** Image showing co-localization of rAAV$_{1/2}$-pOT-mCherry and OT in the PVN. Scale bar = 200 μm. 3v = third ventricle. **c–e** Images of GFP (green, **c**) and mCherry (red, **d**) staining in the vlPAG showing PVN OT fibers surrounding vlPAG GFP neurons (**e**). Scale bar = 300 μm, zoom scale bar = 40 μm. Aq = Aqueduct. **f–i** Retrograde tracing of projections from PVN OT-neurons to the vlPAG. **f** Schema of viral injection showing injection of rAAV$_{1/2}$-pOT-DIO-GFP in the PVN and rAAV$_{1/2}$-CAV2-Cre in the vlPAG of WT rats ($n = 4$ female rats). **g** Image of the PVN from a rat injected with CAV2-Cre into the vlPAG and rAAV$_{1/2}$-OTp-DIO-GFP into PVN, with OT stained in red. White arrows indicate co-localization of GFP and OT. Scale bar = 200 μm. **h**, **i** Magnified insets of the cells indicated by white arrows in the wide field image. Scale bar = 40 μm.

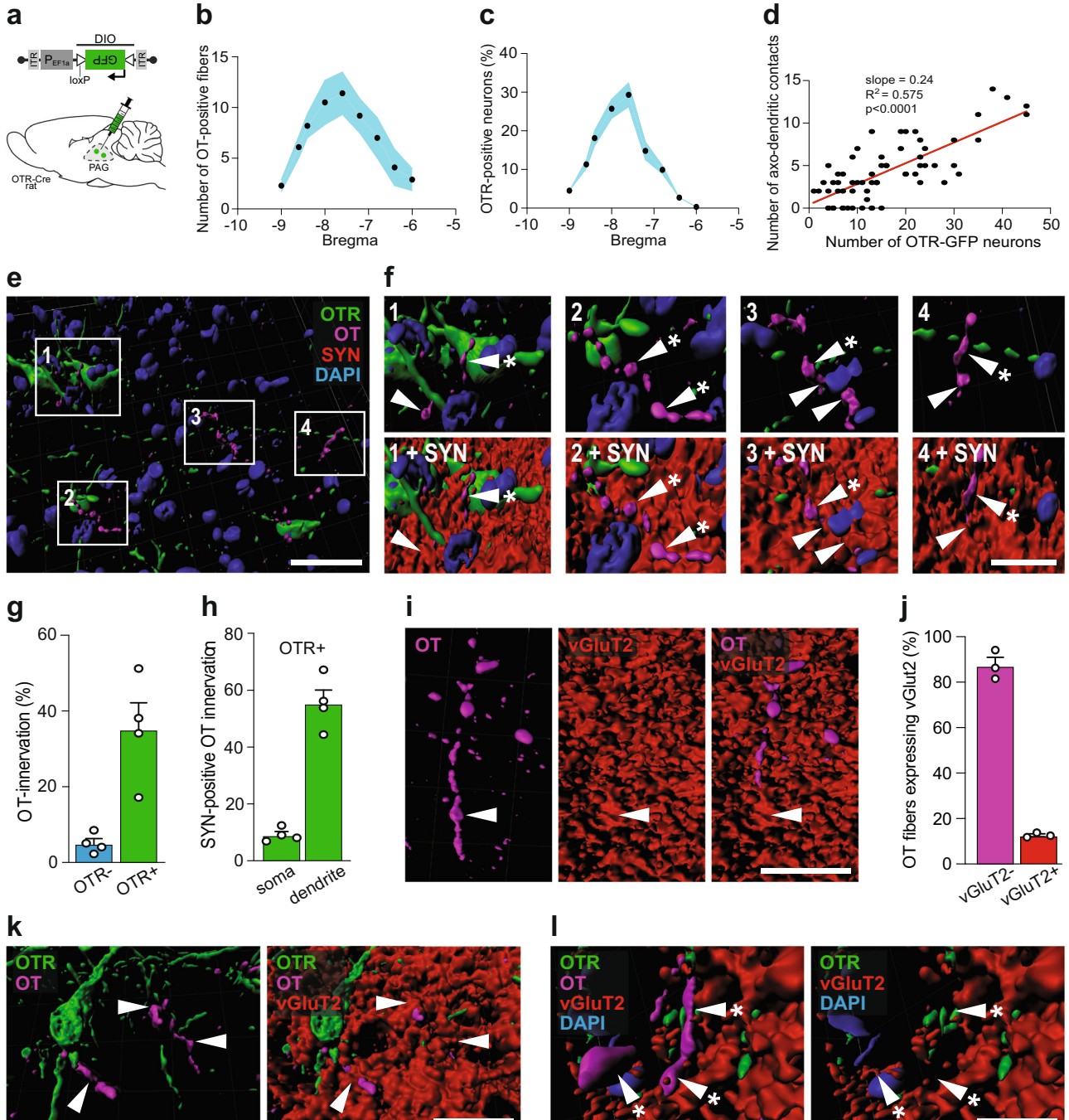

**Fig. 3 | OT fibers form synaptic and non-synaptic contacts with vlPAG neurons of female rats. a** Schema of rAAV$_{1/2}$-pEF1α-DIO-GFP injection in the PAG of OTR-Cre rats. **b** Graph showing the relation between the number of OT fibers in the vlPAG and the bregma level. Blue area represents SEM. n$_{fibers}$ = 290, n$_{rats}$ = 4. **c** Graph showing the relation between the percentage of OTR neurons in the vlPAG and the bregma level. Blue area represents SEM. n$_{cells}$ = 528, n$_{rats}$ = 4. **d** Correlation between the numbers of OTR neurons and OT fibers within the same slice (n$_{slice}$ = 64). Each dot represents one analyzed brain section. Pearson r correlation, R$^2$ = 0.5745, $p$ < 0.0001 (two-sided), slope = 0.2438. **e, f** Three-dimensional reconstruction of OTergic contacts with vlPAG OTR neurons. **e** Overview image showing OTR-neurons (green), OT-fibers (magenta), synaptophysin (SYN, red), and DAPI (blue). **f** Magnified images showing contacts with or without SYN. White arrowheads indicate co-localization of OT (magenta) and SYN (red), while white arrowheads with an asterisk show a mismatch of OT and SYN. DAPI = blue, OTR = green.

**g** Bar graph showing the percentage of OTR positive (n = 496) and negative (n = 3840) cells receiving OT innervation (<1 μm distance between fibers and cells). n$_{rats}$ = 4, 8 images per animal, $p$ = 0.0055 (two-sided). **h** Bar graph showings the percentage of contacts between OT and OTR-positive neurons at somatic and dendritic locations. n$_{rats}$ = 4, 8 images per animal, $p$ < 0.0001 (two-sided). **i** Reconstruction of a vGluT2-positive (red) OT fibers (magenta) within the vlPAG. **j** Bar graph showing that the vast majority of OT fibers within the vlPAG are vGluT2-negative (92.4%). n = 4. **k, l** 3D reconstruction of contacts between an OTR dendrite and OT fibers. **k** Co-localization of OT (magenta) and vGluT2 (red) are indicated by white arrowheads. **l** Mismatch of OT (magenta) and vGluT2 (red) are indicated by white arrowheads with an asterisk. DAPI = blue, OTR = green. n = 4 female rats. Scale bars in order of appearance: 50, 10, 10, 20, and 20 μm. Results are expressed as the mean ± SEM and the individual points of each conditions are represented as white circle. Source data are provided as a Source data file.

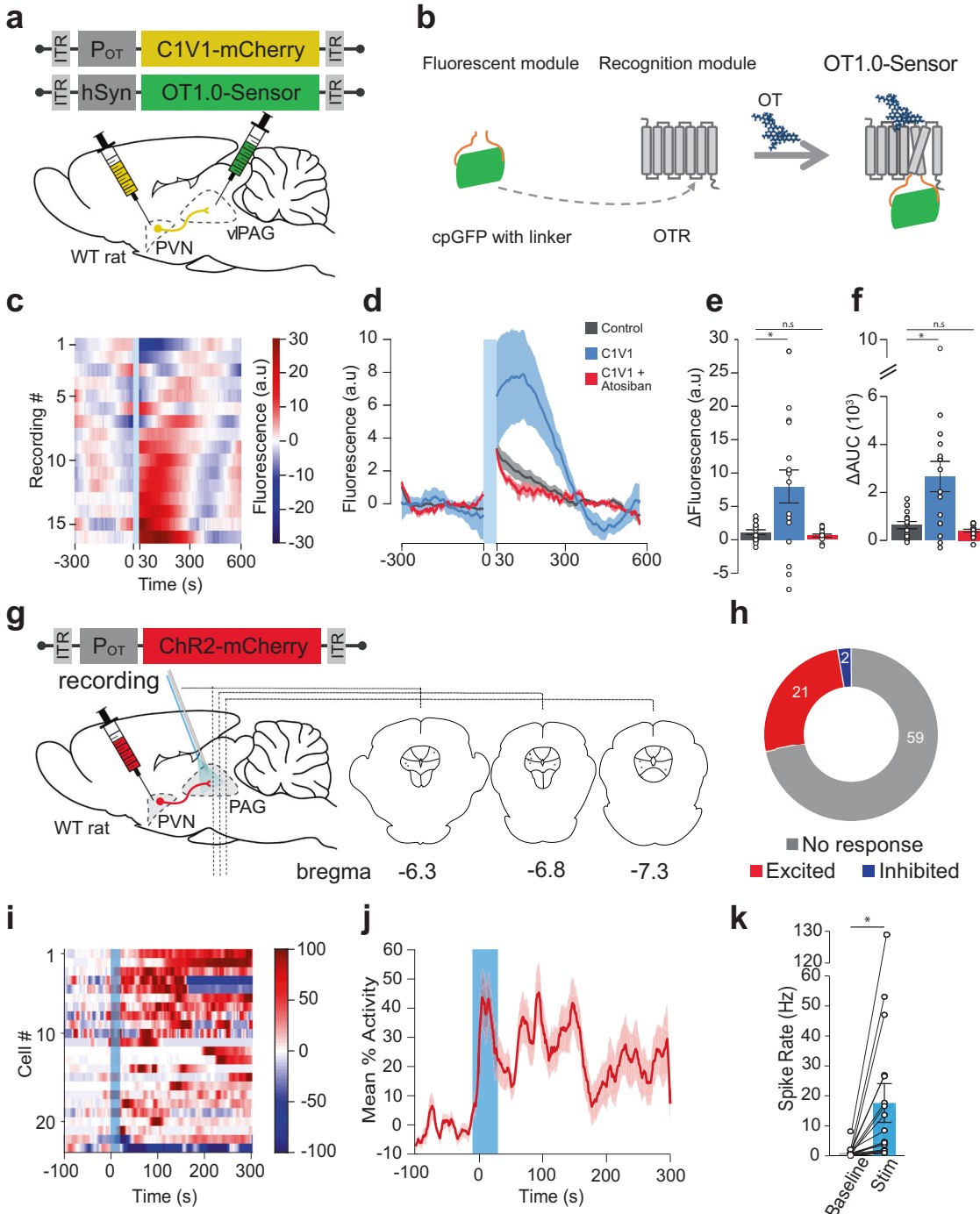

**Fig. 4 | Endogenous OT release in the vlPAG increases vlPAG neuron activity in female rats. a** Schema of injections of rAAV$_{1/2}$-pOT-C1V1-mCherry in the PVN and rAAV$_{1/2}$-hSyn-OT1.0-sensor in the vlPAG. **b** Schema depicting the principle behind the GRAB$_{OT}$ sensor[18]. **c** Fluorescence signal of the C1V1 condition in the vlPAG ($n_{recordings}$ = 16; $n_{slices}$ = 7; $n_{rats}$ = 3). Blue band at 0 s correspond to the yellow stimulation (30 ms pulse at 20 Hz for 30 s). **d** Mean fluorescence (line) ± SEM (shaded) of the control (gray), C1V1 (blue), and C1V1 + atosiban (red) recordings in the vlPAG. **e** Delta of the max fluorescence value between the period before the light stimulation (−300 to 0 s) and the period following the light stimulation (0 to 300 s). The delta of the three conditions are represented. Welch's ANOVA test (W = 6.934, $p$ = 0.0045, $n$ = 16) followed by Dunnett's T3 multiple comparison post hoc test (two-sided): Control vs C1V1: *$p_{adj}$ = 0.0272; Control vs C1V1 + Atosiban: $p_{adj}$: = 0.3133. **f** Delta of the mean Area Under the Curve (AUC) between the period before the light stimulation (−300 to 0 s) and the period following the light stimulation (0 to 300 s). The delta of the three conditions are represented. Welch's ANOVA test (W = 6.934, $p$ = 0.0045, $n$ = 16) followed by Dunnett's T3 multiple

comparison post hoc test (two-sided): Control vs C1V1: *$p_{adj}$ = 0.0118; Control vs C1V1 + atosiban: $p_{adj}$: = 0.2939. **g** Schema of rAAV$_{1/2}$-pOT-ChR2-mCherry injection in the PVN and setup for in vivo electrophysiological recordings (gray electrode), together with blue light (BL) stimulation (blue optic fiber) in the PAG. Recording site is shown on coronal drawings from anterior to posterior. **h** Recorded units' responsiveness. **i** Normalized firing rate of each vlPAG neuron ($n_{cells}$ = 23; $n_{rats}$ = 12) that responded to BL in the vlPAG. 473 nm of BL was added as a 10 ms pulse at 30 Hz for 20 s, 100μW/mm². Dotted lines = BL stimulation. **j** Mean percent activity (line) ± SEM (shaded) calculated from panel (**i**). **k** Difference in mean firing rate between the period before BL (−100 to 0 s) and the maximum activity period following BL stimulation (highest value among moving means with a time window of 21 s, between 0 to +300 s after the start of BL). Paired t-test, *$p$ = 0.0133 (two-sided), $n$ = 17. Results are expressed as the mean ± SEM and the individual points of each conditions are represented as white circle. Source data are provided as a Source data file.

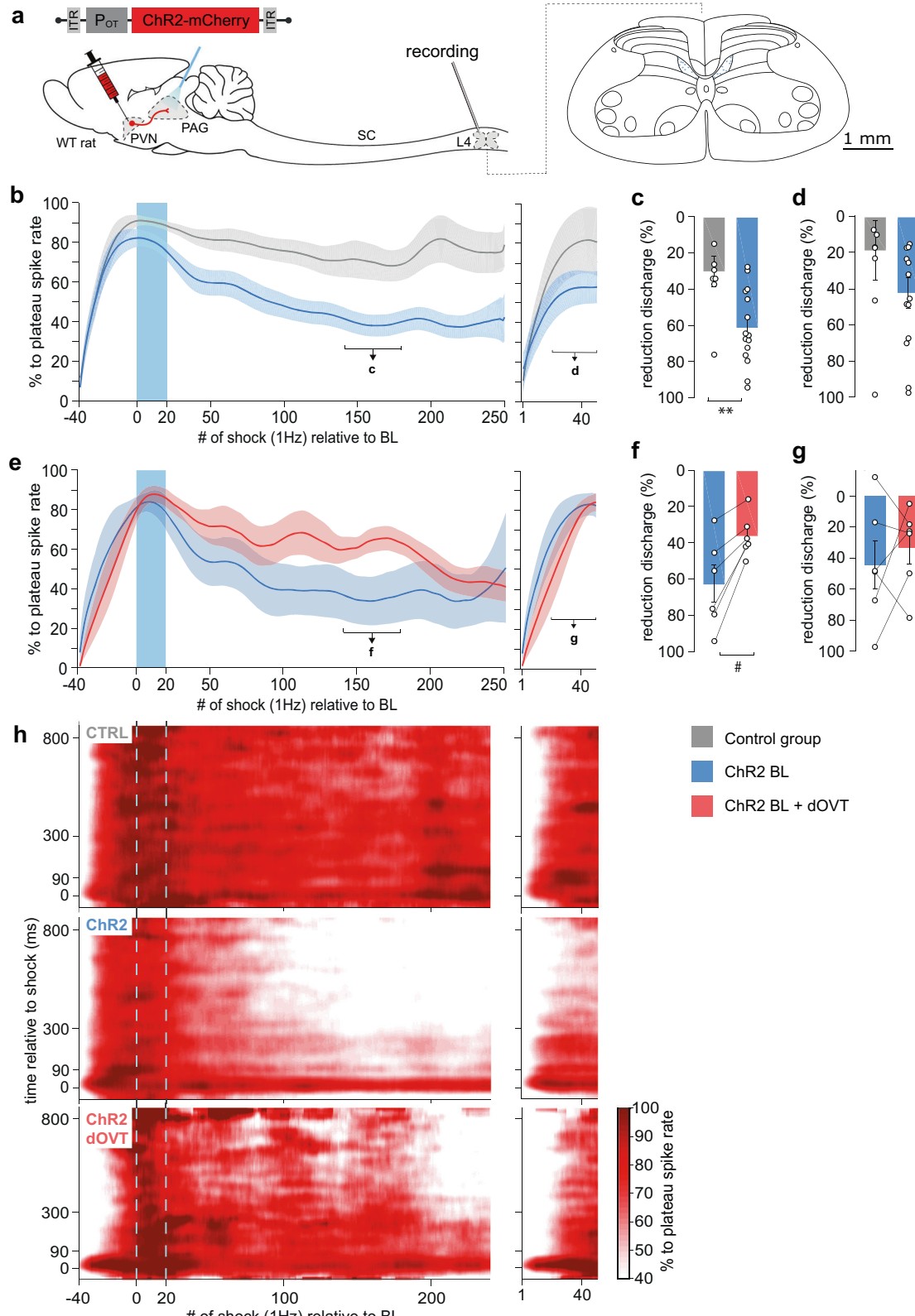

from −40 to 0 s). This WU effect is typically enhanced during pain perception in animals with inflammation[20], which suggests that WU can serve as an index of ongoing nociceptive processing (i.e., a measure of how sensitive the body is to nociceptive stimuli at a given moment). Therefore, we used WU (represented as the percentage of maximal spiking activity following electrical stimulation (1 Hz) to the hind paw receptive field) as our outcome measure for the effect of

vlPAG-BL stimulation on WDR discharge, specifically from primary afferent C-fibers (Supplementary Fig. 9a).

Prior to any vlPAG-BL stimulation, all cells exhibited the maximal WU effect 30 s after the onset of electrical stimulation (Fig. 5b). In control animals (CTRL) that received vlPAG-BL in the absence of ChR2 expression, the WU remained stable up to 250 s after the plateau, despite a gradual reduction over time that was not statistically

**Fig. 5 | Endogenous OT release in vlPAG reduces WDR spinal cord neuronal activity in female rats. a** Schema of rAAV$_{1/2}$-OTp-ChR2-mCherry injection in the PVN and setup for in vivo electrophysiological recordings (gray electrode) of WDR neurons in the rat spinal cord (SC) at the lumbar 4 (L4) level during optogenetic BL stimulation (blue optic fiber) in the vlPAG. Recording sites in layer 5 are shown in the coronal drawing of L4. **b–d** vlPAG BL effect on the spike rate of WDR's C-fiber discharge. **b** Mean time course observed after vlPAG BL in control rats (gray, $n_{cells} = 8$; $n_{rats} = 4$) and OT ChR2-expressing rats (blue, $n_{cells} = 14$ $n_{rats} = 6$). Left and right panels show two consecutive recordings separated by 300 s. Line shadows represent SEM. **c** Percentage of reduction expressed as the minimum level activity observed after a wind-up plateau phase, 140–180 s; Unpaired Wilcoxon rank sum test (two-sided); CTRL ($n_{cells} = 8$) $30.12 \pm 8.60$ vs ChR2 ($n_{cells} = 14$) $61.28 \pm 5.37\%$, $U = 18$, $p = 0.0053$, and **d** 570–600 s after BL onset. Unpaired Wilcoxon rank sum test; CTRL ($n_{cells} = 8$) $18.73 \pm 16.45$ vs ChR2 ($n_{cells} = 14$) $42.23 \pm 8.45\%$, $U = 34$, $p = 0.1450$. **e–g** PAG OTR contribution to the vlPAG BL effect on the spike rate of

WDR's C-fiber discharge. **e** Mean time course observed after vlPAG BL in OT ChR2-expressing rats (blue, $n_{cells} = 6$; $n_{rats} = 4$), and in the same recordings after dOVT injection in the vlPAG (red, $n_{cells} = 6$; $n_{rats} = 4$). Left and right panels show two consecutive recordings separated by 300 s. Line shadows represent SEM. **f** Percentage of reduction expressed as the minimum level activity observed after a wind-up plateau phase, 140–180 s; Paired Wilcoxon signed-rank test (two-sided); ChR2 $63.16 \pm 10.07$ vs dOVT $36.27 \pm 4.8\%$, $W = 0$, #$p = 0.0313$, $n_{cells} = 6$ and **g** 570–600 s after BL onset. Paired Wilcoxon signed-rank test; ChR2 $44.51 \pm 15.61$ vs dOVT $33.16 \pm 10.89\%$, $W = 8$, $p = 0.6875$, $n_{cells} = 6$. **h** Mean smoothed raster plot of WDR discharge level along the relative timing to each single electric shock on the hind paw (vertical axis) and along the accumulating trials of electric shock (horizontal axis), in CTRL animals (top, $n = 8$), ChR2 animals (middle, $n = 14$), and ChR2 animals after dOVT injection in PAG (bottom, $n = 6$). Results are expressed as the mean $\pm$ SEM and the individual points of each conditions are represented as white circle. Source data are provided as a Source data file.

significant. In contrast, vlPAG-BL stimulation in animals expressing ChR2 in OT neurons showed a significant decrease in C-fiber discharge compared to control animals (Fig. 5c, CTRL ($n = 8$) $30.12 \pm 8.60$ vs ChR2 ($n = 15$) $61.28 \pm 5.37$, $p = 0.0053$). This trend was maintained up to 600 s after the end of vlPAG-BL stimulation, as seen in a second series of recordings of the same neurons (Fig. 5d, CTRL ($n = 8$) $18.73 \pm 16.45$ vs ChR2 ($n = 15$) $42.23 \pm 8.45\%$, $p = 0.1450$). A similar effect was found for fast-conducting fibers Aδ- (CTRL ($n = 8$) $34.08 \pm 4.73$ vs ChR2 ($n = 15$) $51.26 \pm 4.1\%$, $p = 0.0337$; Supplementary Fig. 9c–h), but not for non-nociceptive, fast-conducting Aβ- fibers (CTRL ($n = 8$) $24.44 \pm 10.14$ vs ChR2 ($n = 15$) $27.39 \pm 7.91\%$, $p = 0.7763$; Supplementary Fig. 9i–n). While the magnitude of WU reduction was significantly larger in ChR2 animals than CTRL animals, there was no difference in the "inflection" timing of WU dynamics. Specifically, there was no significant difference between CTRL and ChR2 animals for latency (s) to reach the maximum WU (Supplementary Fig. 9b). Importantly, all recorded WDR neurons were impacted by vlPAG-BL, highlighting the effectiveness of this circuit in gating the nociceptive signal at the spinal cord level.

In order to confirm that the recorded effect on WU was due to OT release in the vlPAG, we ran a second series of experiments in which we allowed the WU effect to dissipate over a 10 min interval following the initial stimulation protocols in the ChR2 group. We then infused the specific OTR antagonist, [d(CH2)5,Tyr(Me)2,Orn8]-vasotocin (dOVT), into the vlPAG prior to repeating the same protocol described above. We found that dOVT infusion significantly impaired the vlPAG-BL's ability to reduce WU (ChR2 $63.16 \pm 10.07$ vs dOVT $36.27 \pm 4.80\%$, $p = 0.0313$, $n = 6$; Fig. 5e, f). After the period of maximum WU reduction (from 140 to 180 s), dOVT lost its effectiveness possibly due to diffusion out of the PAG region (ChR2 $44.51 \pm 15.61$ vs dOVT $33.16 \pm 10.89\%$, $p = 0.6875$, $n = 6$; Fig. 5g). The average raster plots shown in Fig. 5h summarize the vlPAG-BL effect in the different groups.

## OT in the vlPAG induces analgesia in both inflammatory and neuropathic pain

Because the vlPAG is known to be a key component of an important descending pain modulatory system, and OT is known to exert an analgesic effect[13], we hypothesized that OTR + neurons in the vlPAG are involved in pain processing. To test this hypothesis, we injected rAAV$_{1/2}$-EF1α-DIO-GFP into the vlPAG of male OTR-IRES-Cre rats. The rats were subdivided into four groups ($n = 7$–8 per group): (1) no manipulation (Control), (2) inflammatory-induced pain sensitization after complete Freund adjuvant injection in the posterior right paw (CFA), (3) acute mechanical nociception (Pain), (4) mechanical nociception 24 h following hind paw CFA injection (CFA + Pain). Rats were euthanized 30 min following each manipulation and brain sections containing vlPAG were collected. We next used c-Fos staining to compare the number of recently activated OTR (GFP+) expressing cells across the groups (Fig. 6a–e). We found that rats exposed to either

painful stimuli or painful stimuli following CFA injection had significantly more activated OTR neurons in the vlPAG ($38.1 \pm 9.2\%$ and $35.9 \pm 3.4\%$, respectively; $p < 0.01$) than rats not exposed to nociceptive stimulation ($18.4 \pm 9.6\%$). There was no statistically significant difference between the CFA group ($26.0 \pm 6.8\%$) and the control group (Fig. 6a).

To verify the functional significance of vlPAG projecting OT neurons in the processing of inflammatory pain-like behaviors, a separate cohort of female wild-type rats received PVN injections of rAAV$_{1/2}$-OTp-ChR2-mCherry (Fig. 6f). We then compared the within-subject effect of optogenetically-evoked OT release in the vlPAG (vlPAG-BL) on mechanical pain-like behavior sensitivity both with and without the presence of CFA-induced inflammatory hyperalgesia (Fig. 6g) and in the chronic constriction injury of the sciatic nerve (CCI) model of neuropathic pain. We found that vlPAG-BL stimulation significantly, but not entirely, alleviated CFA-induced hyperalgesia as indicated by an increase in the mechanical pain-like behaviors threshold from $64.01 \pm 8.059$ g to $120.73 \pm 11.98$ g (mean $\pm$ SEM; Fig. 6g; $p = 0.0019$, $n = 10$). Injection of the blood–brain barrier (BBB)-permeable OTR antagonist, L-368,899, completely blocked the effect of BL in the vlPAG (from $60.5 \pm 4.82$ g to $63.81 \pm 5.12$ g, Fig. 6g; $p = 0.8788$, $n = 10$). After complete wash out of L-368,899, the effect of vlPAG-BL returned to its baseline level (from $86.31 \pm 7.95$ g to $167.29 \pm 14.36$ g; Fig. 6g; $p = 0.0001$, $n = 10$). Finally, we found that vlPAG-BL had no effect on mechanical sensitivity in the absence of any peripheral sensitization when testing the contralateral paw (Supplementary Fig. 10a, d). We next sought to test if this PVN$_{OT}$ → vlPAG$_{OTR}$ circuit is involved in neuropathic pain, given that other pain-related OT pathways fail to affect such symptoms[3,21]. To address this, we performed vlPAG-BL stimulation in the CCI model of neuropathic pain[22]. We found the vlPAG-BL stimulation significantly increases the mechanical pain-like behaviors threshold (mean $\pm$ SEM; $195.68 \pm 27.33$ g to $265.27 \pm 22.26$ g, Fig. 6h; $p = 0.0363$, $n = 7$). Again, the effect of the stimulation is completely blocked by the L-368,899 (from $174.6 \pm 21.32$ g to $181.19 \pm 11.21$ g; Fig. 6h; $p = 0.9478$, $n = 7$). This time, the effect of vlPAG-BL does not significantly return to its baseline level after complete washout of L-368,899 (from $155.49 \pm 16.3$ g to $218.286 \pm 21.75$ g; Fig. 6h; $p = 0.0977$, $n = 7$). Another time, the vlPAG-BL stimulation had no effect when testing the contralateral paw (Supplementary Fig. 10a–c).

To confirm that this effect was driven by vlPAG OTR neurons, we injected male and female OTR-Cre rats with viruses containing either an excitatory chemogenetic receptor, rAAV$_{1/2}$-EF1α-DIO-hm3D(Gq)-mCherry, or a control virus rAAV$_{1/2}$-EF1α-DIO-mCherry ($n = 5$–6 per group) (Fig. 6i, Supplementary Fig. 10d, e). We then repeated the CFA-induced inflammatory hyperalgesia experiments described above and found that chemogenetic excitation of vlPAG OTR neurons by i.p deschlorodozapine (DCZ) induced a significant increase in mechanical pain-like behaviors threshold from $171.27 \pm 5.59$ g to $285.53 \pm 6.36$ g ($p = 0.0006$, $n = 5$–6; Fig. 6j). This effect was not

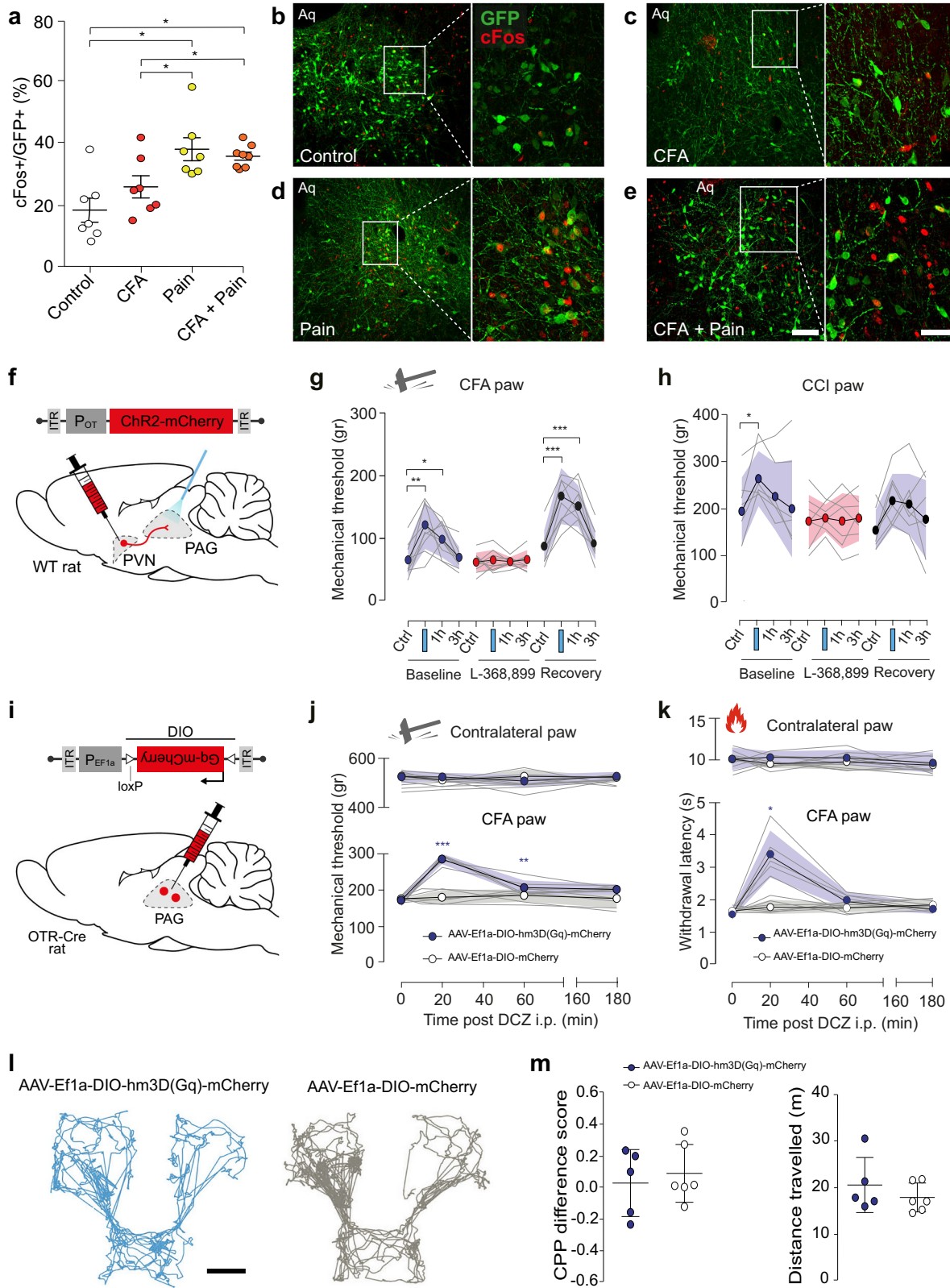

observed in the contralateral paw of the same animals nor in the control virus group that received DCZ injection (Fig. 6j). A similar effect was found in the hot plate test of thermal pain-like behaviors sensitivity, where DCZ increased the latency of response to thermal stimuli from $1.56 \pm 0.06$ s to $3.41 \pm 0.32$ s ($p = 0.0108$, $n = 5$–6; Fig. 6k), this was again not found in the contralateral paw of the same animals nor in the control group (Fig. 6k). Finally, we repeated these

experiments in male rats and found the same results (Supplementary Fig. 10f, g, $n = 6$ per group), ruling out a potential sexual dimorphism of this circuit.

Finally, to understand whether the analgesia was caused by a modulation of the sensory and/or affective component of pain[23], we performed a conditioned place preference (CPP) test using the same cohort from the chemogenetic experiments above (Supplementary

**Fig. 6 | Evoked OT release in vlPAG reduces mechanical hyperalgesia.**
**a** Percentage of c-Fos positive vlPAG OTR neurons under the control condition, painful stimulation, CFA inflammation, and painful stimulation combined with CFA. $n$ = 7–8 per group, Kruskall Wallis test H = 12.01, $p$ = 0.00733, CTRL vs pain, CTRL vs pain + CFA, CFA vs pain, CFA vs pain + CFA $p$ < 0.05, CTRL vs CFA $p$ > 0.05.
**b**–**e** Examples of images showing c-Fos (red) and GFP (green) staining of vlPAG under the different experimental conditions (**b**–**e**). Scale bar = 200 μm, inset scale bar = 75 μm. Aq = aqueduct. **f** Schema of rAAV$_{1/2}$-OTp-ChR2-mCherry injection in the PVN and optic fiber implantation in the PAG. **g**, **h** Threshold of mechanical pain-like behaviors was raised by PAG-BL. The effect of vlPAG-BL was measured at 5 min, 1 h, and 3 h after vlPAG-BL for: **g** the CFA-injected paw, 2-way RM ANOVA test (F$_{interaction}$ = 9.555; $p$ < 0.0001; n$_{rats}$ = 10), followed by multiple comparison post hoc test with Dunnett correction: Baseline Ctrl vs BL: **$p_{adj}$ = 0.0019; Ctrl vs 1 h: *$p_{adj}$ = 0.0269; Recovery, Ctrl vs BL: ***$p_{adj}$ = 0.0001: Ctrl vs 1 h: ***$p_{adj}$ = 0.0005. **h** the CCI-treated paw, 2-way RM ANOVA test (F$_{time}$ = 6.452; $p$ = 0.0012; n$_{rats}$ = 7), followed by multiple comparison post hoc test with Dunnett correction: Baseline Ctrl vs BL:

*$p_{adj}$ = 0.0363. **i** Schema of the injection of rAAV$_{1/2}$-EF1α-DIO-Gq-mCherry in the PAG. **j** Mechanical pain threshold after DCZ administration in the CFA and contralateral paw of female rats expressing Gq-mCherry (blue) or mCherry only (gray) in vlPAG OTR neurons. 2-way RM ANOVA test (F$_{interaction}$ = 21.41; $p$ < 0.0001; n$_{rats}$ = 5–6), followed by multiple comparison post hoc test with Dunnett correction: Gq-mCherry, 0 vs 20: ***$p_{adj}$ = 0.0006; 0 vs 60: **$p_{adj}$: = 0.0089. **k** Thermal pain threshold of female rats expressing Gq-mCherry (red) or mCherry only (gray) in vlPAG OTR neurons after DCZ administration during normal or inflammation (CFA) conditions. 2-way RM ANOVA test (F$_{interaction}$ = 28.29; $p$ < 0.0001; n$_{rats}$ = 5–6), followed by multiple comparison post hoc test with Dunnett correction: Gq-mCherry, 0 vs 20: *$p_{adj}$ = 0.0108. **l** Representative activity traces during the CPP test. Scale bar = 20 cm. **m** Graphs showing the ΔCPP score (left) and total distance traveled (right) for the test and control groups. Unpaired t-test (two-sided): p = 0.6185 (left) and $p$ = 0.3587 (right); $n$ = 6 per group. All results are expressed as average ± SEM and individual animals are represented with the lines, or individual points represented as blue or white circle. Source data are provided as a Source data file.

Fig. 10h). The animals' baseline chamber preference was determined during habituation (see "Methods" for details) and used as the saline-paired control chamber. In contrast, the innately non-preferred chamber was paired with DCZ in order to stimulate vlPAG OTR cells expressing hm3D(Gq) (Fig. 6i). Analysis of the rats' behavior on test day revealed no significant change in preference for the DCZ-paired chamber (Fig. 6l, m), rejecting the assumption of the PVN$_{OT}$ → vlPAG$_{OTR}$ circuit contribution to affective component of pain. Importantly, this was not due to an effect on locomotion as there were no differences between the test and the control group in the total distance travel during the experiment (Fig. 6m).

## Discussion

Here we describe an analgesic pathway recruited by parvOT neurons projecting to the vlPAG (Figs. 1–2), where they release OT to activate GABAergic OTR expressing neurons (Figs. 3–4), which then leads to a decreased response to nociceptive stimuli in spinal cord WDR neurons (Fig. 5). We further showed that activation of this circuit specifically reduces pain-like behaviors (Fig. 6), without alteration of the affective component of pain, in both female and male rats (Fig. 6 and Supplementary Fig. 10).

Past work has shown that OT exerts analgesic effects by acting on various targets of pain-associated areas in the central and peripheral nervous systems[3,24]. The contribution of OT to analgesia is generally attributed to two pathways. The first is an ascending OT pathway that modulates the activity of brain regions processing the affective and cognitive components of pain, such as the amygdala, in which OT alleviates anxiety, especially in the context of chronic pain in rodents[15,17,21]. In humans, OT was found to decrease neural activity in the anterior insula with repeated thermal pain stimulation, thereby facilitating habituation to the cognitive component of the painful stimuli[14]. The second is a descending OT pathway that indirectly promotes analgesia by reducing the activity of SC WDR neurons, which relay nociception in response to painful stimuli[3,25]. While this specific descending OT pathway is effective, it is restricted to inflammatory pain model[3]. In contrast, here we described a powerful descending OT pathway that is effective for both inflammatory and neuropathic pain models across thermal and mechanical modalities.

We identified a population of parvOT neurons that projects to the vlPAG, but not the SON nor the SC. Furthermore, we found that these neurons form synapses with little contribution of glutamate, thus supporting the idea of local axonal delivery of OT, as opposed to volume transmission[26–28]. Of note, we did not decipher the functional involvement of a putative OT/glutamate co-release in this region, a mechanism of general interest for cellular network modulation that remains to be elucidated.

Consistent with previous reports showing that electrical stimulation of the PAG inhibits the firing of dorsal horn neurons in the SC[9]

and generates analgesia[8], we found that nociceptive transmission from C-type primary afferents to WDR neurons in the SC was effectively repressed by endogenous OT release in the vlPAG. Notably, this effect peaked 250 s after OT release and was still observed 10 min after the cessation of blue light. This finding suggests that OT triggers a lasting activation of OTR expressing cells in the vlPAG, which then continually drives the regulation of SC WDR neurons for several minutes after initial OT release. Indeed, in a separate set of experiments, we showed that exogenously applied OTR agonist as well as endogenously evoked OT release excites neurons of the vlPAG and induces analgesia in an inflammatory pain model. Notably, optogenetic release of OT in the vlPAG led to activation of individual neurons at various times, mostly within the first 40 seconds after the onset of blue light stimulation. The neurons' offset timings were also diverse and usually lasted for several minutes after the offset of blue light stimulation. The reason for such variability in the offset times is still unclear. One possibility may stem from the G-protein coupled metabotropic receptor nature of OTRs, which typically produce "slow" post synaptic currents lasting on the order of minutes[29]. Furthermore, although OT axons in the midbrain make synapses[16,30], direct release of OT into the synaptic cleft has never been demonstrated. Thus, it is more likely that OT diffuses from axonal terminals or axonal varicosities *en passant* in the vicinity of OTR neurons[26]. In that case, the action of OT could be synergized across multiple OTR expressing cells, resulting in long-lasting excitation driven by the sum of different active timings. This idea leads to another possible mechanism in which the OT-induced modulation of this pathway relies on the influence of additional non-neuronal cell types within the network, such as astrocytes, as was previously shown in the amygdala[21]. While the specific mechanisms behind the lasting analgesic effect of OT release in the vlPAG are unclear, they would certainly play a critical role in its development as a potential therapeutic target and, therefore, warrant future research.

Importantly, we confirmed the uniqueness of this parvOT pathway by showing that the previously identified parvOT → SC$_{WDR}$ and parvOT → SON pathways do not project to vlPAG. Interestingly, the level of reduction in nociception caused by optogenetic stimulation of parvOT neurons projecting to SON[3] resembled the effect of stimulating vlPAG OT axons. Redundant, parallel circuits that play identical roles in the brain have been previously described (e.g. for feeding behavior[31]). Therefore, the direct projections from parvOT neurons to the SC and the indirect influence of parvOT neurons on sensory WDR neurons via the vlPAG can be interpreted as parallel circuits capable of independently promoting analgesia, particularly in an inflammatory pain model. Although activation of both circuits results in similar electrophysiological inhibition of WDR neurons, they could be triggered by different situations, at different time points, or in different painful contexts. Indeed, we found that the

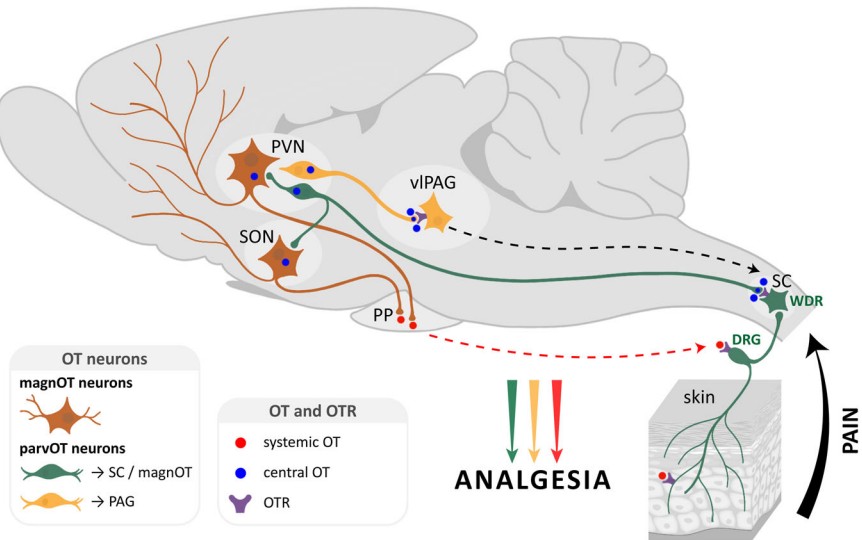

**Fig. 7 | Two distinct ParvOT neuronal populations promote analgesia via release of OT in the vlPAG and in the blood and spinal cord.** We hypothesize that two parallel parvOT pathways are activated by pathological, painful stimuli. Both pathways release OT in various brain regions and the periphery which then leads to a reduction in nociception.

PVN$_{OT}$ → vlPAG$_{OTR}$ circuit promotes analgesia in the chronic neuropathic pain model condition and in both male and female rats, whereas previous work found that the parvOT → SC$_{WDR}$ circuit does not[3]. Moreover, considering that the vlPAG is an important area for the regulation of various defensive behaviors[32], and that OT can mediate defensive behaviors, it will be important for future research to determine if vlPAG OTR neurons might be involved in other functions beyond nociception.

**Local vlPAG GABAergic circuit recruited by OT**
It has been previously shown that electric stimulation of PAG inhibits the firing of dorsal horn neurons in SC[9], resulting in analgesia[8]. However, although these studies did not explore the role of neuronal inputs to the PAG such as the OTergic ones. Because this analgesic effect was interrupted by lesions of the RVM[33], the RVM is currently considered an essential link between PAG and SC. Furthermore, analgesic projections from the PAG to the RVM are glutamatergic[34] and tonically inhibited by local PAG GABAergic neurons[35]. Interestingly, local OT infusion increases the spontaneous activity of PAG neurons[12]. In the current study, we confirm this by showing that endogenous OT release in PAG triggered not only excitation of some neurons whose spontaneous spike rates were low, but also caused inhibition of other neurons whose spontaneous spike rates were high (Fig. 4i). We also show that OTR neurons in the vlPAG are GABAergic and that TGOT application increases their electrical activity. Thus, an interesting hypothesis could be that the OT-excited neurons are local GABAergic interneurons[35] while the OT-inhibited neurons might correspond to the glutamatergic projection neurons[34]. Stimulating the endogenous release of OT leads then to the increase of GABAergic interneurons, decreasing the activity of the glutamatergic projection neurons, decreasing also RVM activity, and promoting analgesia. However, further investigations are needed to determine the exact nature of those OT-modulated neuronal populations.

**vlPAG OTR neurons reduce pain sensation, but not its affective component**
We further found that painful stimulation increased c-Fos levels in vlPAG OTR neurons, indicating that these neurons are endogenously recruited in the context of pain processing. These findings confirm and expand on previous work demonstrating that PAG OTR neurons in mice express c-Fos after noxious stimulation[36].

By using our newly generated OTR-IRES-Cre transgenic rats, we were able to specifically activate OTR neurons in the vlPAG. We found that this activation led to a decrease in nociception in both inflammatory and neuropathic pain models, but failed to alter place preference. This suggests the circuit we dissected here is not involved in the affective, memory component of pain[23]. Interestingly, the opposite effect of OT on affective valence, but not on physical pain-like behaviors has been demonstrated in the central nucleus of amygdala[21]. In addition, OT was reported to modulate pain-like behaviors at the level of the insular cortex by enhancing GABAergic transmission and causing downstream effects leading to reduced nociceptive signaling in the spinal cord[37]. Taken together, these findings further emphasis the sheer variety of effects mediated by OT and highlight the need for continued efforts to dissect the precise anatomical and functional characteristics of the central OT system.

**Sex (absence of) difference in OT vlPAG induced analgesia**
Interestingly, clinical literature, as well as some animal research[38], mention a sex relevance in OT-induced analgesia. However, one might consider that clinical studies mostly rely on intra-nasal or intra-venous administration of exogenous OT, thus flooding any brain-area specific effect of OT, while most of sex differences reported in animal studies at the level of the spinal cord were performed via release of endogenous OT. In the present study, we mainly focus on female animals, extensively demonstrating the analgesic action of the specific PVN$_{OT}$ → vlPAG$_{OTR}$ circuit in both inflammatory and neuropathic pain conditions. Interestingly, this analgesic effect was confirmed in male animals (Supplementary Fig. 10), pointing to similar analgesic mechanisms of the PVN$_{OT}$ → vlPAG$_{OTR}$ circuit regardless of sex differences[39]. This is in accordance with recent anatomical analysis that reveal limited, if any, differences of OT neuros projections between female and male rodents[40]. Given that increasing evidence points toward major sex differences in pain processing, results from our group and others on OT circuits highlight an important, well-conserved OTergic analgesic mechanism.

In conclusion, we identified a subpopulation of parvocellular OT neurons that mediate analgesia by recruiting the PAG-controlled descending pain modulatory system (Fig. 7). This study further describes and supports the role of OT as an analgesic molecule and points to the OTR as a potential therapeutic target. To this end, we generated the

OTR-IRES-Cre line of rat, which will greatly enhance our ability to research this therapeutically relevant receptor. Finally, it should be noted that the inconsistent results regarding sex differences and subjective pain ratings[41] found in human clinical studies of OT effects on pain[24] may be due to the limitations of intranasal OT administration[42]. Therefore, future research should be oriented toward developing synthetic OT agonists with the ability to cross the blood–brain barrier more efficiently than OT itself[43–46].

## Methods

### Animals

Adult female (all figures) and male (Supplementary Figs. 4 and 10) Wistar wild type and Wistar OTR-IRES-Cre rats (>8 weeks old; 250−350 g; Chronobiotron, Strasbourg, France) were used for this study. All animals were tattooed, sexed and genotyped (Kapa2G Robust HotStart PCR Kit, Kapa Biosystems; Hoffman La Roche) one week after birth. Animals were housed by sex, in groups of three under standard conditions (room temperature, 22 °C; 12/12 h light/dark cycle) with ad libitum access to food, water and behavioral enrichment. All animals that underwent behavioral testing were handled and habituated to the experimenter two weeks before stereotaxic surgery. After one week of post-surgical recovery, the rats were habituated to the applicable behavioral testing room and handling routines for an additional two weeks prior to the start of experiments. All behavioral tests were conducted during the light period (i.e., between 7:00 and 19:00). All experiments were conducted in accordance with European law, under French Ministery license 3668-2016011815445431 and 15541-2018061412017327, and German Animal Ethics Committee of the Baden Württemberg license G-102/17.

### Viral cloning and packaging

Recombinant Adeno-associated virus (serotype 1/2) carrying either a conserved region of the OT promoter or EF1α promoter and genes of interest in direct or "DIO" orientations were cloned and produced as reported previously[15]. Briefly, HEK293T cells (#240073, Addgene, USA) were used for viral production. rAAV produced included: rAAV$_{1/2}$-OTp-mCherry(or Venus), rAAV$_{1/2}$-OTp-ChR2-mCherry, rAAV$_{1/2}$-EF1α-DIO-GFP (or mCherry), and rAAV$_{1/2}$-EF1α-DIO-hM3Dq-mCherry, rAAV$_{1/2}$-OTp-C1v1-mCherry, rAAV$_{1/2}$-hSyn-OT1.0. The canine adenovirus serotype 2 (CAV2-CMV-Cre) was purchased from the Institute of Molecular Genetics in Montpellier CNRS, France. rAAV genomic titers were determined with QuickTiter AAV Quantitation Kit (Cell Biolabs, Inc., San Diego, California, USA) and RT-PCR using the ABI 7700 cycler (Applied Biosystems, California, USA). rAAV titers were between $10^9$ and $10^{10}$ genomic copies/µl.

### Stereotaxic injections

All surgeries were performed on rats anesthetized with 2.5% isoflurane and receiving Bupivacaine (s.c., 2 mg/kg) or carprofen (i.p., 5 mg/kg) and lidocaine applied locally[47]. rAAV were injected into the PVN, SON, and vlPAG in different combinations, as needed by each experiment, and allowed to express for four weeks. The coordinates were chosen using the Paxinos rat brain atlas[48] (PVN: ML: +/−0.3 mm; AP: −1.4 mm; DV: −8.0 mm; SON: ML: +/−1.8 mm; AP: −1.2 mm; DV: −9.25 mm; PAG: ML: +/−0.5 mm; AP: −7.0 mm; DV: −5.9/−5.0 mm). Each site was injected with a total of 300 nL of viral solution via a glass pipette at a rate of 150 nl/min using a syringe pump. Verification of injection and implantation sites, as well as expression of genes of interest were confirmed in all rats post hoc (see "Histology" section). Rhodamine conjugated Retrobeads (Lumafluor Inc., Durham, NC, USA) were diluted 1:10 with 1x PBS and injected at a volume of 150 nl. Spinal cord Retrobeads injections was performed during the same surgery as virus injection (see "In vivo extracellular recording of WDR SC neurons" for details on the spinal cord surgery).

## Generation of OTR-IRES-Cre rats

**Cloning of the rat OXTR-Cre targeting vector.** The OXTR-Cre targeting vector was cloned by modifying the plasmid Snap25-IRES2-Cre (Allen Institute for Brain Science[49]) The final vector contained the IRES2-Cre sequences, followed by a bovine growth hormone poly-adenylation site, and homology arms for targeted integration of the oxytocin receptor locus, comprised of 1.3 kb and 1.4 kb genomic sequences. Homology arms were generated by PCR on genomic DNA from Sprague Dawley rats using the following primer pairs: OXTR_fwd_upper (5′-GTCGACAGAAAACTGGTGGGTTTGCC-3′) together with OXTR_rev_upper (5′-GCTGCTAGCGAAGACTGGAGTC CACACCACC-3′) and OXTR_fwd_lower (5′-ACCCGGGAATTCTGTG-CATGAAGCTGCATTAGG-3′) together with OXTR_rev_lower (5′-TAG TTTAAACGTGCATTCGTGTATGTTGTCTATCC-3′). The upper homology arm was inserted using the restriction enzymes SalI and NheI, while the lower arm was inserted using XmaI and PmeI. Vector sequences can be obtained upon request.

**Design of gRNAs and functional testing.** We used the online tool CRISPOR (http://crispor.org) for selection of the guide RNA (gRNA) target sites in the OXTR gene[50]. Three gRNA target sites were chosen with high specificity scores (>83[51]) for binding in the OXTR 3′ UTR site where we aimed to introduce the IRES-Cre coding sequences.

For identification of the most effective gRNA, dual expression vectors based on px330 were cloned, harboring an expression cassette for the selected gRNAs and Cas9. In addition, the OXTR 3′ UTR gRNA target regions were inserted into a nuclease reporter plasmid (pTAL-Rep[52]) in between a partly duplicated, nonfunctional β-galactosidase gene. Hela cells were transfected with a combination of one of the px330 plasmids and the OXTR-specific reporter vector. After transfection, the Cas9-nuclease-induced double-strand breaks stimulated the repair of the lacZ gene segments into a functional reporter gene, the activity of which was determined in cell lysates using an o-nitro-phenyl-β-D-galactopyranosid (ONPG) assay. A luciferase expression vector was also added to the transfection mix and luciferase activity was measured for normalization. The most effective gRNA target site including PAM was determined as 5′-ACTCCAGTCTTCCCCC GTGGTGG-3′.

**Specificity of the CRISPR/Cas induced genomic modification.** The CRISPOR program was used to identify potential off-target sites for the selected OXTR gRNA (5′-ACTCCAGTCTTCCCCCGTGGTGG-3′). No off-target sites were detected in an exonic sequence or on the same chromosome. In addition, only two potential target sites harboring at least four mismatches in the 12 bp adjacent to the PAM could be identified by the software.

**Generation of transgenic rats.** Sprague Dawley (SD) rats (Charles River) were bred in standard cages (Tecniplast) under a 12-h light/dark cycle in a temperature-controlled environment with free access to food and water at the Central Institute of Mental Health, Mannheim. All animal protocols were approved by the Regierungs-präsidium Karlsruhe. SD single-cell embryos were injected using standard microinjection procedure[53]. In brief, microinjections were performed in the cytoplasm and male pronuclei of zygotes with a mixture of Cas9 mRNA (10 ng·/µl), sgRNA expression vector (6 ng/µL) and the OXTR-Cre targeting vector (2 ng/µl) as the repair substrate. The injected embryos were cultured in M2 Medium at 37 °C in 5% $CO_2$ and 95% humidified air until the time of injection. Surviving oocytes were transferred to the oviducts of pseudo pregnant Sprague Dawley rats. Transgenic animals were identified by polymerase chain reaction of tail DNA (DNeasy kit, Qiagen, Hilden, Germany) using primers for Cre[54].

DNA of Cre-positive animals was further used for detection of homologous recombination at the OXTR locus. For this purpose, the

targeted region was amplified by PCR using the Q5 polymerase (NEB) with 100−200 ng of genomic DNA as a template. Primers were selected which bind both up and downstream of the insertion site (outside of the homology arms), and were each combined with a primer located within the IRES2-Cre construct. For the 5' insertion site, the primers OXTR_check (5'-CAGCAAGAAGAGCAACTCATCC-3') together with Cre_rev (5'-CATCACTCGTTGCATCGAC-3') and, for the 3' site, CRISPR_bGH_fwd (5'-GACAATAGCAGGCATGCTGG-3') together with OXTR_rev_check (5' AGCCAGGTGTCCAAGAGTCC-3') were used.

## Western blot for the detection of Cre protein in OTR-IRES-Cre rats

Total protein lysates were isolated from frozen rat tissue samples using RIPA buffer with Halt Protease- und Phosphatase-Inhibitor-Cocktail (100X) (ThermoFisher Scientific, Waltham, MA, USA) Concentration was measured using Pierce™ BCA Protein Assay Kit (ThermoFisher Scientific, Waltham, MA, USA). Proteins (40−60 µg) were separated on a NuPAGE 4−12% w/v Bis-Tris 1.0-mm minigel (Invitrogen, Carlsbad, CA) and detected by primary antibody anti-Cre polyclonal rabbit Antibody (NB100-56135) according to the manufacturer's instructions (dilution 1:1000). An anti-Gapdh mouse monoclonal antibody was used as a control (dilution 1:5000). IRDye 800CW and IRDye 680 (LICOR Bioscience, Lincoln, NE, USA) were used as secondary antibodies (Dilution 1:10.000). Protein bands were visualized using the Odyssey Infrared Imaging System and quantified using Image Studio Lite 4.0 software (both LI-COR Biosciences, Lincoln, NE, USA).

## Histology

After transcardial perfusion with 4% paraformaldehyde (PFA) and post fixation overnight, brain sections (50 µm) were collected by vibratome slicing and immunohistochemistry was performed as previously described[15]. The list of primary antibodies used is available in the key resource table. For secondary antibodies, signal was enhanced by Alexa488-conjugated IgGs (1:1000) or CY3-conjugated or CY5-conjugated antibodies (1:500; Jackson Immuno-Research Laboratories). All images were acquired on a confocal Leica TCS microscope. Digitized images were processed with Fiji and analyzed using Adobe Photoshop. For the visualization of OTergic axonal projections within the PAG, we analyzed brain sections ranging from bregma −6.0 to −8.4 mm.

## RNAScope in situ hybridization

RNAScope reagents (Advanced Cell Diagnostics, Inc., PN320881) and probes (OTR: 483671-C2 RNAscope Probe and Cre: 312281 RNAscope Probe) were used to detect the presence of specific mRNA expression using in situ hybridization. Brains were processed as described above using nuclease-free PBS, water, PBS, and sucrose. We followed the manufacturer's protocol with a few modifications: (1) immediately after cryosectioning, slices were washed in nuclease-free PBS to remove residual sucrose and OCT compound. (2) Hydrogen peroxide treatment was performed with free-floating sections prior to slice mounting. (3) Sections were mounted in nuclease-free PBS at room temperature. (4) Pretreatment with Protease III was performed for 20 min at room temperature. (5) No target retrieval step was performed.

To determine the percentage of PAG neurons co-expressing Cre and OTR mRNA, in situ hybridization using the RNAscope™ HiPlex Assay (Advanced Cell Diagnostics (ACD), Hayward, CA, USA; in a version for AF488, Atto550 and Atto647 detection) was performed. All procedures were conducted using fresh frozen 16 µm brain sections (3 PAG-containing slices per brain). During the initial hybridization step Cre (CRE-T2, cat. no. 312281-T2, ACD) and OTR (Rn-Oxtr-T6, cat. no. 483671-T6, ACD) probes were applied. Images were acquired and processed using an Axio Imager M2 fluorescent microscope (Zeiss, Gottingen, Germany) with an automatic z-stage and Axiocam 503 mono camera (Zeiss), and subsequently with Zen (3.1 blue edition and

3.0 SR black edition, Zeiss), CorelDraw 2020 (Corel Corporation, Ottawa, Canada), ImageJ and HiPlex Image Registration Software v1.0 (ACD). OTR and Cre mRNA-expressing cells in the PAG region from one brain hemisphere per section, were counted with an ImageJ Cell Counter plugin. Neurons were identified by the presence of a DAPI-stained nucleus and/or an unambiguous cell-like distribution of fluorescent mRNA dots.

## Three-dimensional reconstruction and analysis of OT-OTR contacts in the PAG

Confocal images were obtained using a Zeiss LSM 780 confocal microscope (1024 × 1024 pixel, 16-bit depth, pixel size 0.63-micron, zoom 0.7). For the three-dimensional reconstruction, 40-µm-thick z-stacks were acquired using 1 µm-steps. Imaris-assisted reconstruction was performed as previously described[21,47,55]. In brief, surface reconstructions were created based on the four individual channels (DAPI, OT, OTR-GFP, SYN/vGlut2). Co-localization of OT signal with SYN or vGlut2 was confirmed both manually and through the association/overlap function of IHC-labeled puncta in the Imaris software. IHC intensity of vGlut2 and SYN were assessed by creating spheres that precisely engulfed somata or dendrites as previously described[47].

## Ex vivo imaging of oxytocin endogenous release

**Slice preparation.** To confirm the endogenous release of oxytocin following the light stimulation of the PVN fibers, 5−6 weeks old females Wistar HAN rats ($n = 3$) received injections of the OT1-sensor virus into the vlPAG and the C1V1 virus into the PVN. Following a 2 weeks recovery period, rats were anesthetized by administering i.p. ketamine (Imalgene 300 mg/kg) and paxman (Rompun, 60 mg/kg). Transcardial perfusions were performed using an ice-cold, NMDG-based aCSF was used containing (in mM): NMDG (93), KCl (2.5), $NaH_2PO_4$ (1.25), $NaHCO_3$ (30), $MgSO_4$ (10), $CaCl_2$ (0.5), HEPES (20), D-Glucose (25), L-ascorbic acid (5), Thiourea (2), Sodium pyruvate (3), N-acetyl-L-cysteine (10) and Kynurenic acid (2.5). The pH was adjusted to 7.3−7.4 using HCl, after bubbling in a gas comprised of 95% $O_2$ and 5% $CO_2$. Rats were then decapitated, brains were removed and 350 µm thick coronal slices containing the vlPAG were obtained using a Leica VT1000s vibratome. Slices were placed in a recuperation chamber filled with normal aCSF at room temperature for at least 1 h. Normal aCSF was composed of (in mM): NaCl (124), KCl (2.5), $NaH_2PO_4$ (1.25), $NaHCO_3$ (26), $MgSO_4$ (2), $CaCl_2$ (2), D-Glucose (15), at pH 7.3−7.4 and continuously bubbled in 95% $O_2$−5% $CO_2$ gas. Osmolarity of all aCSF solutions were controlled to be between 290−310 mOsm. Finally, slices were transferred from the holding chamber to an immersion-recording chamber and superfused at a rate of 2 ml/min.

**OT1.0-sensor imaging.** The spinning disk confocal microscope used to perform OT1.0-sensor imaging was composed of a Zeiss Axio examiner microscope with a ×40 water immersion objective (numerical aperture of 1.0), mounted with a X-Light Confocal Unit−CRESTOPT spinning disk. Images were acquired at 2 Hz with an optiMOS sCMOS camera (Qimaging). The fluorescent focal planes were illuminated for 30 ms (OGB1: 475 nm) in bright-field mode using a Spectra 7 LUMENCOR. The different hardware elements were synchronized through the MetaFluor 7.8.8.0 software (Molecular Devices). The fibers were stimulated using a 575 nm wavelength illuminated for 30 ms at 20 Hz for 30 s after 5 min of a baseline period. Using the Fiji software, the brightness and contrast parameters were automatically adjusted. As no cell could be clearly identified, the variations of fluorescence were measured on the whole plane recorded. Further offline data analysis was performed using a custom-written Python-based script available on the editorial website. Briefly, an inverted exponential fit was estimated to compensate the photobleaching of the recordings. Recordings in which the stimulation-induced artifact disrupted the fitting were discarded. Endogenous release of oxytocin was estimated as

changes in the fluorescence signal expressed as a change in the maximum value of fluorescence or in the Area Under the Curve (AUC) after the light stimulation. For this, a delta of the maximum value of fluorescence and the AUC is obtained by subtracting the value measured during the 300 s period after the stimulation to the value obtained during the 300 s period before the stimulation. Slices were incubated for at least 1 h in 1 µM atosiban before being recorded. All recordings were performed at room temperature (25 °C).

### Ex vivo patch-clamp recording of vlPAG-OTR neurons

**Slice preparation.** To validate the functionality of vlPAG OTR neurons in OTR-IRES-Cre rats using electrophysiology, 12–13-week-old female OTR-IRES-Cre rats ($n = 4$) received injections of the Cre-dependent reporter virus, $rAAV_{1/2}$-pEF1α-DIO-GFP, into the vlPAG. Following a 4–8 week recovery period, rats were anesthetized by administering i.p. ketamine (Imalgene 300 mg/kg) and paxman (Rompun, 60 mg/kg). Transcardial perfusions were performed using an ice-cold, NMDG-based aCSF was used containing (in mM): NMDG (93), KCl (2.5), $NaH_2PO_4$ (1.25), $NaHCO_3$ (30), $MgSO_4$ (10), $CaCl_2$ (0.5), HEPES (20), D-Glucose (25), L-ascorbic acid (5), Thiourea (2), Sodium pyruvate (3), N-acetyl-L-cysteine (10), and Kynurenic acid (2). The pH was adjusted to 7.4 using either NaOH or HCl, after bubbling in a gas comprised of 95% $O_2$ and 5% $CO_2$. Rats were then decapitated, brains were removed and 350 µm thick coronal slices containing the hypothalamus were obtained using a Leica VT1000s vibratome. Slices were warmed for 10 min in 35 °C NMDG aCSF then placed in a room temperature holding chamber filled with normal aCSF for at least 1 h. Normal aCSF was composed of (in mM): NaCl (124), KCl (2.5), $NaH_2PO_4$ (1.25), $NaHCO_3$ (26), $MgSO_4$ (2), $CaCl_2$ (2), D-Glucose (15), adjusted to pH 7.4 with HCL or NaOH and continuously bubbled in 95%-$O_2$ 5%-$CO_2$ gas. Osmolarity of all aCSF solutions were maintained between 290 and 310 mOsm/L. Finally, slices were transferred from the holding chamber to an immersion-recording chamber and superfused at a rate of 2 ml/min.

**Patch clamp recording.** We targeted GFP+ or GFP- neurons in the vlPAG for whole-cell patch-clamp recording. The recording pipettes were visually guided by infrared oblique light video microscopy (DM-LFS; Leica). We used 4–9 MΩ borosilicate pipettes filled with a K-gluconate-based solution composed of (in mM): $KMeSO_4$ (135), NaCl (8), HEPES (10), $ATPNa_2$ (2), GTPNa (0.3). The pH was adjusted to 7.3–7.4 with KOH and osmolality was adjusted with sucrose to 290–310 mOsm/L, as needed. Data were acquired with an Axopatch 200B (Axon Instruments) amplifier and digitized with a Digidata 1440 A (Molecular Devices, CA, USA). Data were sampled at 20 kHz and lowpass filtered at 5 kHz using the pClamp10 software (Axon Instruments). Further analysis was performed using Clampfit 10.7 (Molecular Devices; CA, USA) and Mini analysis 6 software (Synaptosoft, NJ, USA) in a semi-automated fashion (automatic detection of events with chosen parameters followed by a visual validation). First spike latency quantification is measured as the duration preceding the first spike of a neuron submitted to a super-threshold stimulus of 50 pA.

**TGOT stimulation.** Finally, to validate the functionality of the putative OTR-Cre expressing cells in the vlPAG, we recorded GFP + neurons in gap free (current clamp mode). Following a 5 min baseline recording period, a solution containing the OTR agonist, [Thr⁴Gly⁷]-oxytocin (TGOT, 0.4 µM), was pumped into the bath for 20 s. The recording continued for a total of 20 min while the frequency of action potentials (APs) was quantified as the measure of neuronal activity. As a control, we repeated this procedure while patching GFP-neurons in the same vicinity. Neurons were held at -50 mV throughout the recording and all ex vivo experiments were conducted at room temperature.

### In vivo extracellular recording of vlPAG neurons

To test the effects of endogenous OT release on OTR cells of the vlPAG in vivo, female Wistar rats ($n = 11$) were injected with $rAAV_{1/2}$-pOT-ChR2-mCherry into the PVN. After a 4–8 week recovery period, rats were anaesthetized with 4% isoflurane and placed in a stereotaxic frame before reducing the isoflurane level to 2%. A silicone tetrode coupled with an optical fiber (Neuronexus, USA) was inserted into the PAG to allow for stimulation of the ChR2-expressing axons of PVN-OT neurons projecting to vlPAG while recording the activity of putative OTR expressing neurons in the vicinity. Optical stimulation was delivered using a blue laser (λ 473 nm, output of 3 mW, Dream Lasers, Shanghai, China) for 20 s at 1 Hz, with a pulse width of 5 ms. Extracellular neuronal activity was recorded using a silicone tetrode coupled with an optic fiber (Q1x1-tet-10mm-121-OAQ4LP; Neuronexus, USA). Data were acquired on an MC Rack recording unit (Multi Channel Systems), and spikes were sorted by Wave Clus[56]. Spike data was analyzed with custom MATLAB (MathWorks) scripts and the MLIB toolbox (Stüttgen Maik, Matlab Central File Exchange).

The firing rate of each recording unit was smoothed by convolution of Gaussian distribution, whose width was 10 s and standard deviation was 5. The mean firing rate of the baseline period (BSmean) was defined as 0% activity and subtracted from the firing rates (FR) of the whole period (FR- BSmean). Maximum absolute activity (max(FR-BSmean)) was found using the highest absolute value among moving means of (FR- BSmean) with a time window of 21 s. When maximum absolute activity was found to exceed the BSmean, it was defined as 100% activity (cell#1-21), whereas when maximum absolute activity was found to be below BSmean, it was defined as −100% activity (cell#22-23).

### In vivo extracellular recording of WDR SC neurons

Adult female Wistar rats ($n = 22$) were anesthetized with 4% isoflurane and a then maintained at 2% after being placed in a stereotaxic frame. A laminectomy was performed to expose the L4-L5 SC segments, which were then fixed in place by two clamps positioned on the apophysis of the rostral and caudal intact vertebras. The dura matter was then removed. To record wide-dynamic-range neurons (WDR), a silicone tetrode (Q1x1-tet-5mm-121-Q4; Neuronexus, USA) was lowered into the medial part of the dorsal horn of the SC, at a depth of around 500–1100 µm from the dorsal surface (see Fig. 5a for localization of recorded WDRs). We recorded WDR neurons of lamina V as they received both noxious and non-noxious stimulus information from the ipsilateral hind paw.

We measured the action potentials of WDR neurons triggered by electrical stimulation of the hind paw. Such stimulation induced the activation of primary fibers, whose identities can be distinguished by their spike onset following each electrical stimuli (Aβ-fibers at 0–20 ms, Aδ-fibers at 20–90 ms, C-fibers at 90–300 ms and C-fiber post discharge at 300 to 800 ms). When the WDR peripheral tactile receptive fields are stimulated with an intensity corresponding to 3 times the C-fiber threshold (1 ms pulse duration, frequency 1 Hz), a short-term potentiation effect, known as wind-up (WU), occurs that leads to an increased firing rate of WDR neurons[57,58]. Because the value of WU intensity was highly variable among recorded neurons within and across animals, we averaged the raster plots two dimensionally across neurons within each group of rats. We further normalized these data so that the plateau phase of the maximal WU effect was represented as 100 percent activity. As WU is dependent on C-fiber activation, it can be used as a tool to assess nociceptive information in the SC and, in our case, the anti-nociceptive properties of OT acting in the vlPAG. We recorded WDR neuronal activity using the following protocol: 40 s of hind paw electric stimulation to induce maximal WU followed by continued electrical stimulation to maintain WU while simultaneously delivering 20 s of vlPAG blue light stimulation (30 Hz, 10 ms pulse width, output -3 mW), followed by another 230 s of

electrical stimulation alone to observe the indirect effects of OT on WU in WDR neurons. Electrical stimulation was ceased after the 290 s recording session to allow the WU effect to dissipate. Following a 300 s period of no stimulation, the ability of the WU effect to recover was assessed by resuming electrical stimulation of the hind paw for 60 s of WU. After another 10 min period without stimulation, we sought to confirm the effects of vlPAG OT activity on WU intensity by injecting 600 nl of the OTR antagonist, dOVT, (d(CH2)5-Tyr(Me)-[Orn8]-vaso-tocine; 1 μM, Bachem, Germany) into the vlPAG of the rats expressing ChR2, and repeated the protocol described above.

The spikes of each recording unit were collected as raster plots with the vertical axis showing the time relative to electric shock, and the horizontal axis showing the number of electric shocks. Next, the raster plots were smoothed by convolution of the Gaussian distribu-tion (horizontal width = 100 ms, vertical width = 20 ms, standard deviation = 20. The total number of C fiber derived spikes occurring between 90 and 800 ms after each electric shock was counted. The spike counts were smoothed with a moving average window of 21 s and the window containing the maximal activity was defined as '100% activity', which was then used to normalize the activity of each recording unit. Finally, the normalized percent activity from each recording unit was averaged for each experimental condition and plotted in Fig. 5.

### Determination of the phase of estrous cycle

To ensure consistency across studies (Eliava et al., 2016), all in vivo electrophysiological recordings were conducted during the diestrus phase of the ovarian cycle, which we determined using vaginal smear cytology[59]. Briefly, a micropipette filled with 100 μL of saline solution (NaCl 0.9%) was placed in the rats' vagina and cells were dissociated by pipetting up and down at least three times. A drop of the smear was placed on a glass slide and observed using a light microscope with a 40x or 100x objective lens. Ovarian phase was determined based on the proportion of leukocytes, nucleated epithelial cells, and anucleate cornified cells. Animals in metestus, proestrus and estrus phases were excluded from experiments and reintroduced once they reached diestrus.

### Behavioral tests

**Optogenetics.** For in vivo optogenetic behavioral experiments, we used a blue laser (λ 473 nm, 100 mW/mm², DreamLasers, Shanghai, China) coupled to optical fiber patch cables (BFL37-200-CUSTOM, EndA=FC/PC, EndB=Ceramic Ferrule; ThorLabs, USA). Optical fiber probes (CFMC12L10, Thorlabs, USA) were bilaterally implanted into the vlPAG (Coordinates relative to bregma: ML = ± 2.0 mm, AP = −6.7 mm, DV = −7.0 mm, medio-lateral angle=10°) under isoflurane anesthesia (4% induction, 2% maintenance), and then stabilized with dental cement. Following a two-week recovery period, all ani-mals received special handling to habituate them to the fiber con-nection routine. Optical stimulation to the vlPAG was delivered using a series of pulse trains (intensity = ~3 mW, frequency = 30 Hz, pulse width = 10 ms, duration = 20 s) during all applicable behavioral experiments.

**Chronic constriction of the sciatic nerve.** To produce the model of chronic neuropathic pain, we surgically implanted a cuff around the sciatic nerve to induce a chronic constriction injury as previously described[60]. Briefly, under isoflurane anesthesia (5%) via a facemask, an incision was made 3–4 mm below the femur on the right hind limb and 10 mm of the sciatic nerve was exposed. A sterile cuff (2 mm section of split PE-20 polyethylene tubing; 0.38 mm ID/1.09 mm OD) was posi-tioned and then closed around the sciatic nerve. The skin was then sutured shut to enclose the cuff and allow chronic constriction to occur.

**Mechanical hyperalgesia.** Mechanical sensitivity was assessed using calibrated digital forceps (Bioseb®, France) as previously described[20]. Briefly, the habituated rat is loosely restrained with a towel masking the eyes in order to limit environmental stressors. The tips of the forceps are placed at each side of the paw and gradual force is applied. The pressure required to produce a withdrawal of the paw was used as the nociceptive threshold value. This manipulation was performed three times for each hind paw and the values were averaged. After each trial, the device was cleaned with a disinfectant (Surfa'Safe, Anios laboratory®).

**Inflammatory hyperalgesia.** In order to induce peripheral inflamma-tion, 100 μL of complete Freund adjuvant (CFA; Sigma, St. Louis, MO), was injected into the right hind paw of the rat. All CFA injections were performed under light isoflurane anesthesia (3%). Edema was quanti-fied by using a caliper to measure the width of the dorsoplantar aspect of the hind paw before and after the injection of CFA. In effort to reduce the number of animals used, we did not include an NaCl-injected group, as it has already been shown that the contralateral hind paw sensitivity is not altered by CFA injection[45].

**Thermal hyperalgesia.** To test the animal thermal pain sensitivity threshold, we used the Plantar test with the Hargreaves method (Ugo Basile®, Comerio, Italy) to compare the response of each hind paw of animals having received unilateral intraplantar CFA injection. The habituated rat is placed in a small box and we wait until the animal is calmed. We then exposed the hind paw to radiant heat and the latency time of paw withdrawal was measured. This manipulation was per-formed three times for each hind paw and the values were averaged. After each trial, the device was cleaned with a disinfectant (Surfa'Safe, Anios laboratory).

**Conditioned place preference.** The device is composed of two opaque conditioning boxes (rats: 30 × 32 cm; mice: 22 × 22 cm) and one clear neutral box (30 × 20 cm). Animals were tracked using a video-tracking system (Anymaze, Stoelting Europe, Ireland) and apparatus was cleaned with disinfectant (Surfa'Safe, Anios labora-tory) after each trial. The animals underwent CPP as previously described[21]. Briefly, all rats underwent a three-day habituation period during which they were able to freely explore the entire apparatus for 30 min. On the 3rd habituation day, exploration behavior was recorded for 15 min to determine the animals' innate side preference. On the 4th day, animals were injected with saline (i.p, 1 mL/kg) and placed in their innately preferred chamber (unpaired box) for 15 min. Four hours later, animals were injected with DCZ (i.p, 100 μg/kg at 1 mL/kg[61]) to stimulate vlPAG OTR neurons expressing hM3D(Gq), and then placed in the innately non-preferred chamber (paired box) for 15 min. On the 5th day, the animals were placed in the neutral chamber and allowed to explore the entire apparatus for 15 min. To control for potential locomotor effects, the total distance traveled during the test period was quantified and compared between all groups.

**c-Fos expression after painful stimulation.** For the c-Fos experiment (Fig. 6), rats were pinched on the right hind paw (with or without CFA) three times using the same forceps as for the mechanical pain threshold procedure. Progressive pressure was applied until the rat exhibited pain-like behaviors by either retracting its paw or squeaking. The control groups were handled in the same way but no pressure was applied. Rats were perfused 90 min later as described previously.

### Drugs

All drugs used in this study are listed in the reagents and resource table.

| REAGENT OR RESOURCE | SOURCE | IDENTIFIER |
|---|---|---|
| **Antibodies** | | |
| Chicken anti-GFP primary antibody | Abcam | ab13970 |
| Mouse NeuN primary antibody | Millipore | MAB377 |
| Rabbit vGluT2 primary antibody | Synaptic systems | 135 103 |
| Rabbit anti-dsRed primary antibody | Living Colors | 632496 |
| GAD67 | Millipore | MAB5406 |
| Guinea-pig anti-Fluorogold primary antibody | Protos Biotech Corp | NM-101 |
| Mouse c-Fos polyclonal primary antibody | Santa-Cruz | sc-8047 |
| Mouse monoclonal anti-OT primary antibody | Provided by Dr. Harold Gainer | PS 38 |
| DAPI | Vector Laboratories | H-1200-10 |
| Synaptophysin | Abcam | Ab32127 |
| Rabbit polyclonal anti-vGluT2 primary antibody | SYSY | 135403 |
| **Bacterial and virus strains** | | |
| rAAV1/2-OTp-Venus | Lab made | N/A |
| rAAV1/2-OTp-mCherry | Lab made | N/A |
| rAAV1/2-OTp-FLEX-GFP | Lab made | N/A |
| CAV2-CMVp-Cre | Montpellier Vectorology Platform | N/A |
| rAAV1/2-OTp-ChR2-mCherry | Lab made | N/A |
| rAAV1/2-OTp-C1V1-mCherry | Lab made | N/A |
| rAAV1/2-EF1αp-FLEX-GFP | Lab made | N/A |
| rAAV1/2-EF1αp-FLEX-mCherry | Lab made | N/A |
| rAAV1/2-EF1αp-FLEX-hM3D(Gq)-mCherry | Lab made | N/A |
| rAAV2/9-hSyn-OT1.0-sensor | Lab made | N/A |
| **Chemicals, peptides, and recombinant proteins** | | |
| Fluorogold™ | Santa-Cruz | sc-358883 |
| Oxytocin Receptor Antagonist | Merck | L-368,899 |
| Deschloroclozapine | MedChemExpress | 1977-07-7 |
| TGOT | Bachem® | 50-260-164 |
| CFA | Sigma | 32160405 |
| dOVT | Bachem | d(CH2)5-Tyr(Me)-[Orn8]-vasotocine |
| Retrobeads™ | Lumafluor | N/A |
| **RNAscope reagents and materials** | | |
| RNAscope HiPlex12 detection Reagents (488, 550, 647) | ACD | 324106 |
| RNAscope HiPlex Probe Diluent | ACD | 324301 |
| RNAscope Wash Buffer Reagents | ACD | 310091 |
| RNAscope Protease III & IV Reagents | ACD | 322340 |
| RNAscope HiPlex Cleaving Stock Solution | ACD | 324136 |
| RNAscope HiPlex CRE-T2 Probe | ACD | 312281-T2 |
| RNAscope HiPlex Rn-Oxtr-T6 Probe | ACD | 483671-T6 |
| RNAscope HiPlex12 Positive Control Probe – Rn | ACD | 324331 |
| RNAscope HiPlex12 Negative Control Probe | ACD | 324341 |
| UltraPure 20X SSC Buffer | invitrogen | 15557-044 |
| ProLong Gold antifade reagent | invitrogen | P36930 |
| ImmEdge Hydrophobic Barrier Pen | VECTOR LABORATORIES | H-40000 |
| Tween 20 | Sigma-Aldrich | P1379 |
| **Experimental Models: Organisms/Strains** | | |
| Rattus Norvegicus (Wistar HAN) | Janvier | N/A |
| Rattus Norvegicus (Sprague Dawleys) | Charles River | N/A |
| OTR-IRES-Cre (Wistar HAN) | Lab made | N/A |
| **Software and Algorithms** | | |
| Graphpad prism 7.05 | www.graphpad.com | N/A |
| Fiji | www.imagej.net/Fiji | N/A |
| Adobe Photoshop CS5 | www.adobe.com | N/A |
| Adobe Illustrator 16.05 | www.adobe.com | N/A |
| CorelDraw 2020 | www.coreldraw.com | N/A |
| HiPlex Image Registration Software v1.0 | ACD | 300065 |
| Matlab | Mathworks | N/A |
| Python | Open-source | N/A |
| Zen (3.1 blue edition and 3.0 SR black edition) | Zeiss | N/A |
| **Other** | | |
| Optic fiber implants | www.thorlabs.com | CFMC12L10 |
| 473 nm Blue Laser Generator | www.dreamlasers.com | SDL-473-XXXT |
| Programmable Pulse Stimulator (A.M.P.I.) | www.ampi.co.il | Master-9 |

## Statistics and reproducibility

All individual observations (Cre expression, anatomical qualitative observations) were repeated at least 5 times.

For ex vivo electrophysiology data, a nonparametric Friedman test followed by Dunn's post hoc multiple comparisons test was used to compare the mean AP frequencies across the three conditions (Baseline vs TGOT vs wash) (Supplementary Fig. 1). A two-tailed paired t-test was used to test the difference of the first spike latency before and after TGOT application (Fig. 1). Differences were considered significant at $p < 0.05$.

For ex vivo imaging of OT release, a Welch's ANOVA test followed by a Dunnett's T3 multiple comparison was used to compare the change in the fluorescence (max fluorescence and AUC) after stimulating the C1V1 (with or without atosiban) to the change observed in the control condition (Fig. 4). The analysis of the recordings were performed using a custom-written Python-based script available on the editorial website. Statistical tests were then performed with GraphPad Prism 7.05 (GraphPad Software, San Diego, California, USA). Differences were considered significant at $p < 0.05$.

For in vivo electrophysiology data analysis, a paired-sample t-test was used to compare the average spike rates between the baseline and peak activity of PAG neurons in response to BL stimulation (Fig. 4). A nonparametric, unpaired Wilcoxon rank sum test was used to compare the reduction in discharge of SC neurons between the wild type and the ChR2-expressing animals (Fig. 5). A paired Wilcoxon signed-rank test was used to compare the reduction discharge of SC neurons in the ChR2-expressing animals, between the "without dOVT" and "with dOVT" conditions (Fig. 5). Unpaired Wilcoxon rank sum tests were used to compare the latencies to reach the maximum, minimum, and the half value between the wild type and the ChR2-expressing rats (Supplementary Fig. 9).

A Kruskall Wallis test with Dunn's post hoc test was used to compare the percent co-localization of c-Fos+ and GFP + cells across

pain conditions (Fig. 6). Behavioral data are expressed as mean ± standard deviation (SD). Statistical tests were performed with Graph-Pad Prism 7.05 (GraphPad Software, San Diego, California, USA) and MATLAB. For CFA and CCI (Fig. 6 and Supplementary Fig. 10) animals, a two-way ANOVA with repeated measure followed by multiple comparisons post hoc tests with Dunnett correction was used. The multiple comparisons were done within each condition (baseline, antagonist, recovery) by comparing each time point (blue light, 1 h, 3 h) to its own control. The same statistical procedure was used for CFA animals after DCZ administration (Fig. 6 and Supplementary Fig. 10). The ΔCPP score was calculated with the following formula in order to control for time spent in the neutral chamber: ΔCPP score = (paired$_{postcond}$ − unpaired$_{postcond}$)−(paired$_{hab}$−unpaired$_{hab}$). An unpaired two-tailed t-test was used to compare the effect of DCZ on CPP difference score across the two groups as well as the distance traveled by the two groups (Fig. 6). All rats with off-target viral injection sites were removed from analysis. Differences were considered significant at $p < 0.05$.

Asterisks are used to indicate the significance level: *$0.01 \le p < 0.05$, **$0.001 \le p < 0.01$, ***$p < 0.001$.

### Reporting summary
Further information on research design is available in the Nature Portfolio Reporting Summary linked to this article.

## Data availability
The raw data generated in this study have been deposited in the Zenodo database under accession code 10.5281/zenodo.7473865. The data generated in this study are provided in the Source data file. In addition, all data that support the findings of this study are available from the corresponding authors upon request. Source data are provided with this paper.

## Code availability
Python code for ex vivo GRAB analysis can be found at the following address: https://github.com/Etienneclcr/GRAB_PAG; Matlab code for in vivo electrophysiological recordings can be found at the following address: https://github.com/Etienneclcr/ephy-in-vivo-PAG.

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

## Acknowledgements

This work was supported by the Centre National de la Recherche Scientifique contract UPR3212, the Université de Strasbourg contract UPR3212; the University of Strasbourg Institute for Advanced Study (USIAS) fellowship 2014-15, Fyssen Foundation research grant 2015, NARSAD Young Investigator Grant 24821, Agence Nationale de la Recherche (ANR, French Research Foundation) grants n° 19-CE16-0011-0, 19-CE37-0019, and 20-CE18-0031 (to A.C.); the Graduate School of Pain EURIDOL, ANR-17-EURE-0022 (to A.C. and E.C.C.); ANR-DFG grant GR 3619/701, PHC PROCOPE and PICS07882 grants (to A.C. and V.G.); Deutsche Forschungsgemeinschaft (DFG, German Research Foundation) grants GR 3619/15-1, GR 3619/16-1 (to V.G.); SFB Consortium 1158-2 (to V.G., S.H., and B.D.); French Japanese governments fellowship B-16012 JM/NH and Subsidy from Nukada Institute for Medical and Biological Research (to M.I.); Fyssen Foundation fellowship (to A.L.); Région Grand Est fellowship (to D.K.); DFG Postdoc Fellowship AL 2466/1-1 (to F.A.); the Foundation of Prader-Willi Research post-doctoral fellowship (to C.P.S and F.A.); DAAD Postdoc Short term research grant 57552337 (to R.P.); DFG Walter Benjamin Position – Projektnummer 459051339 (to Q.K.). National Heart, Lung, and Blood Institute Grant NIH HL090948, National Institute of Neurological Disorders and Stroke Grant NIH NS094640, and funding provided by the Center for Neuroinflammation and Cardiometabolic Diseases (CNCD) at Georgia State University (to J.E.S.). The authors thank Prof. Yulong Li for providing the GRAB_OTR plasmid, Drs. Romain Goutagny and Vincent Douchamps for in vivo electrophysiology advices, the Chronobiotron UMS 3415 for all animal care and the technical plateau ComptOpt UPR 3212 for behavior technical assistance.

## Author contributions

Conceptualization: A.C.; methodology: A.B., A.C., A.L., C.P.S., D.K., E.C.C., F.A., J.E.S., L.H., M.I., M.M., O.Ł., Q.K., and V.G.; in situ hybridization: A.G., A.K., A.T., and F.A.; immunohistochemistry: A.L., F.A., M.K.K., and S.K.; ex vivo patch-clamp electrophysiology: D.K., E.C.C., H.P., and L.H.; in vivo electrophysiology: M.I., M.M., and O.Ł.; GRAB monitoring: E.C.C.; behavior: L.H. and M.I.; transgenic rats line generation: K.S. and D.B.; transgenic rats line validation: A.K., H.F., J.S., L.D., and M.W.; writing: A.C., A.L., B.D., E.C.C., F.A., J.E.S., M.I., M.K.K., O.Ł., P.D., R.P., S.H., S.K., and V.G.; funding acquisition: A.C. and V.G.; supervision: A.C. and V.G.; project administration: A.C.

## Competing interests

The authors declare no competing interests.

## Additional information

[1]Centre National de la Recherche Scientifique and University of Strasbourg, Institute of Cellular and Integrative Neuroscience, 67000 Strasbourg, France. [2]Department of Neuropeptide Research in Psychiatry, Central Institute of Mental Health, University of Heidelberg, Mannheim 68159, Germany. [3]Center for Neuroinflammation and Cardiometabolic Diseases, Georgia State University, Atlanta, USA. [4]Institute of Human Genetics, University Hospital Heidelberg, Heidelberg, Germany. [5]Van Andel Institute, Grand Rapids, MI, USA. [6]Department of General Psychiatry, Center of Psychosocial Medicine, University of Heidelberg, 69115 Heidelberg, Germany. [7]Institute of Medical Psychology, Heidelberg University Hospital, 69115 Heidelberg, Germany. [8]Ruprecht-Karls University Heidelberg, Heidelberg, Germany. [9]Department of Molecular Biology, Central Institute of Mental Health, University of Heidelberg, Mannheim 68159, Germany. [10]Department of Neurophysiology and Chronobiology, Institute of Zoology and Biomedical Research, Faculty of Biology, Jagiellonian University, Krakow 30-387, Poland. [11]Present address: Cortical Systems and Behavior Laboratory, University of California, San Diego, La Jolla, CA 92093, USA. [12]These authors contributed equally: Mai Iwasaki, Arthur Lefevre, Ferdinand Althammer, Etienne Clauss Creusot. [13]These authors jointly supervised this work: Valery Grinevich, Alexandre Charlet. ✉e-mail: valery.grinevich@zi-mannheim.de; acharlet@unistra.fr

