## [Peer Review File · Nature Communications]

An analgesic pathway from parvocellular oxytocin neurons to the periaqueductal gray in ratsREVIEWER COMMENTS

Reviewer #1 (Remarks to the Author):

Alexandre Charlet et al's manuscript, which titled "A novel analgesic pathway from parvocellular oxytocin neurons to the periaqueductal gray", is a worthy article. The authors used a newly generated line of transgenic rats (OTR-IRES Cre) to determine that most of the vIPAG OTR expressing cells being targeted by OT projections are GABAergic in nature. The results showed that both optogenetically-evoked axonal OT release in the vIPAG as well as chemogenetic activation of OTR vIPAG neurons results in a long-lasting overall increase of vIPAG neuronal activity. It identified a new subpopulation of parvocellular OT neurons that mediate analgesia by recruiting the PAG-controlled descending pain modulatory system. However, the manuscript conclusion has been involved in previous studies, although the used methods are different. For example, in "Abstract", the manuscript described "This then leads to an indirect suppression of sensory neuron activity in the spinal cord and strong analgesia. Finally, we describe a novel OT-vIPAG-spinal cord circuit that seems critical for analgesia in the context of both inflammatory and neuropathic pain". But there are a few content for the spinal cord study. It needs to revise.

Also, the authors should revise the following questions.

1. The clinical studies showed that the analgesic effect of intranasal OT administration was different between genders. In Mai Iwasaki et al's study, adult female and male Wistar wild type and Sprague Dawley OTR-IRES-Cre rats (>8 weeks old; 250 - 350 g; Chronobiotron, Strasbourg, France) were used. It was wondered if the authors consider the sex difference in OT analgesia effect.
2. In "Materials and Methods, Stereotaxic injections", the manuscript described "rAAVs were injected into the PVN, SON and vIPAG in different combinations, as needed by each experiment, and allowed to express for four weeks. The coordinates were chosen using the Paxinos rat brain atlas 41 (PVN: ML: +/-0.3 mm; AP: -1.4 mm; DV: -8.0 mm; SON: ML: +/-1.8 mm; AP: -1.2 mm; DV: -9.25 mm; PAG: ML: +/-0.5 mm; AP: -7.0 mm; DV: -5.9/-5.0 mm)". The methods were not easy, especially SON injection. The author should have provided anatomical proof.
3. Also, the manuscript described "Each site was injected with a total of 300 nL of viral solution via a glass pipette at a rate of 150 nl/min using a syringe pump". I think that this injection process takes time. How to ensure the animal behaviors do not affect the brain nucleus injection?

Reviewer #2 (Remarks to the Author):

The authors showed that a subset of oxytocin neurons project to the ventrolateral PAG and induce excitation of the majority of oxytocin receptor-expressing neurons in the ventrolateral PAG, leading to analgesia after inflammation or after chronic constriction of the sciatic nerve.

Oxytocin has been shown to induce analgesia via acting on the dorsal root ganglion, spinal cord, PAG and insular cortex. This study further demonstrates that PAG oxytocin receptor-dependent analgesia is induced by activation of a new subset of oxytocin neurons. This is an interesting study, which will attract readers in a variety of fields. Following points should be considered.

1. Major finding of this study is demonstration of a novel subset of oxytocin neurons. Thus clear description of the distribution pattern of PVN oxytocin neurons projecting to the PAG as compared to neurons projecting to the spinal cord is helpful for readers. Is it possible to show a distribution map?
2. This study demonstrate analgesic actions of PVN-PAG oxytocin system, In this study photic or chemogenic activation induced analgesia. Are there any evidences suggesting physiological functions of the oxytocin system? The authors showed that painful stimuli activate oxytocin receptor-expressing neurons in the PAG. Why the oxytocin receptor antagonist had no effects in response to pain stimuli when no BL stimuli were applied?
3. Details of painful stimuli in the experiments of Fos should be described.
4. Majority of experiments were performed in female rats. Is there any reason?
5. Is there no influence of the phase of estrous cycle for behavioral pain response?
6. A new transgenic rat line, OTR-IRES-Cre line was produced in this study. The rat line appears to be a knock-in line. However, a CRISPR/CAS method might have produced mis-integration. Was

proper genomic insertion confirmed by performing DNA sequencing of PCR products amplified by a combination of a primer for upstream of the inserted site and a primer of downstream of the inserted site?

7. The number of rats used for detection of Cre mRNA and OTR mRNA should be described clearly.
8. All PVN-PAG oxytocin projections are ipsilateral? Information of laterality should be described clearly. Is there any difference in anatomy or physiology between left and right sides? How many rats received left side injection or right side injection for anatomical or behavioral experiments?
9. Activity of a small number of PAG OTR-expressing neurons was suppressed by stimulation of oxytocin neurons. Both excitatory and inhibitory actions were blocked by an oxytocin receptor antagonist? Possible explanation of no response or inhibitory response is helpful for readers.
10. The expression "evoked OT release in vPAG" was used in the paper. However, OT release was not directly examined. Stimulation of OT fibers does not necessarily mean OT release. Precise expression should be used.
11. Is there any evidence that blue light application to the PAG did not activate passing fibers projecting to the spinal cord?
12. In the experiments with an oxytocin receptor antagonist injection into the PAG, vehicle was injected for the control group (Fig 5 C1)? If so, in figure 5 C1 "ChR2 BL" should read "ChR2 BL + Vehicle".
13. "we found that vPAG-BL had no effect on mechanical sensitivity .. when testing the contralateral paw (Figure 6D2)" "mechanical stimuli 195.7... to 265.3 (Figure S6A1-2)..... Figure S6A2: n=7)." Figure numbers are correct?

Reviewer #3 (Remarks to the Author):

In this manuscript, Iwasaki et al. investigate the role of oxytocin (OCT) expressing cells in the paraventricular (PVN) nuclei in the descending analgesic pathway. For this, they generated rats for which Cre-expression was driven by production of the oxytocin receptor (OCT-Cre rats). Using ex-vivo slice electrophysiology, they confirm that OCT-expressing neurons in PAG (labeled with GFP via AAV-DIO injection) increase firing in response to the selective OTR agonist TGOT. They used anterograde and retrograde techniques to show that OT neurons in PVT consist of distinct populations sending projections to the ventrolateral periaqueductal gray (vPAG) and the spinal cord (SC). Optogenetic activation of PVN-OT axons in vivo within the vPAG led to increased activity of approximately 26% of neurons recorded within the vPAG. More in depth in-vivo studies reveal optogenetic activation of PVN-OT axons within the vPAG could reduce wide dynamic range (WDR) neuronal responses evoked by electrical stimulation of the hind paw. Lastly, Iwasaki et al. shows that this same stimulation of PVN-OT axons in the vPAG can reduce sensory pain behavior in models of inflammatory and neuropathic pain; an effect which they report is sensitive to the OTR antagonist L-368,899. Overall, this exciting and novel study provides new insight into a unique OT pathway that can modulate noxious input via the vPAG. This is important for the development of novel analgesics that could target OT circuits. However, there are a number of concerns that limit enthusiasm, which are discussed below:

1. The representative ephys trace in Figure SE1 clearly shows that TGOT increases firing frequency in response to a depolarizing step current, yet this is not mentioned. The authors just focus on latency to first spike, which may be biologically difficult to alter given that baseline latency is relatively rapid.
2. It is unclear why the authors did not use ex-vivo slice electrophysiology to test the effects of ChR2+ OT+ axons on recorded neurons in the vPAG. This would allow for the identification of OTR+ neurons using TGOT, and the use of OTR antagonists or glutamate antagonists to confirm that any changes to excitability were due to release of OT from ChR2+ axons. This would also confirm that OT can be released via ChR2+ stimulation as the effects seen in Figure 4 could be due to release of glutamate (not OT) from the small population of VGlut2+ OT axons. Figure 5 addresses some of this, but using dOVT in this paradigm would confirm the cellular mechanism.

3. In regards to the previous comment, it is a bit of a stretch to conclude the BL-evoked OT release leads to an overall excitation of putative OTR+ vIPAG neurons.

4. The explanation of Figure 6D2 in the text does not match what is presented in the Figure. The figure shows what appears to be an experiment using the chronic constriction injury, while the text is referring to absence of vIPAG-BL effect in the paw contralateral to CFA injection. This is the same for Figure S6A1-A2.

5. The intermittent use of male and female rats is not well justified. While it seems a majority of rats used were female, it is unclear why males were introduced only in some of the experiments. Furthermore, sex-specific differences were not explored nor mentioned, which is of interest considering reported sex differences in clinical OT studies.

6. Statistical analyses is rather vaguely defined. Mention of tests used for analysis in Figure legends including t and F values would be more informative.

Answers to Reviewers' Comments

We are most grateful to all reviewers for their time and effort in evaluating our manuscript. We thank them for expressing their thoughtful and constructive concerns, which we addressed point-by-point below. We hope that the reviewers will find our responses satisfactory, allowing us to publish this study. For the reviewers' convenience, our responses here as well as the new data in the revised manuscript are typed in blue.

Reviewer #1:

Alexandre Charlet et al's manuscript, which titled "A novel analgesic pathway from parvocellular oxytocin neurons to the periaqueductal gray", is a worthy article. The authors used a newly generated line of transgenic rats (OTR-IRES Cre) to determine that most of the vIPAG OTR expressing cells being targeted by OT projections are GABAergic in nature. The results showed that both optogenetically-evoked axonal OT release in the vIPAG as well as chemogenetic activation of OTR vIPAG neurons results in a long-lasting overall increase of vIPAG neuronal activity. It identified a new subpopulation of parvocellular OT neurons that mediate analgesia by recruiting the PAG-controlled descending pain modulatory system. However, the manuscript conclusion has been involved in previous studies, although the used methods are different. For example, in "Abstract", the manuscript described "This then leads to an indirect suppression of sensory neuron activity in the spinal cord and strong analgesia. Finally, we describe a novel OT-vIPAG-spinal cord circuit that seems critical for analgesia in the context of both inflammatory and neuropathic pain". But there are a few content for the spinal cord study. It needs to revise.

We thank the reviewer for this critical point. Accordingly, we modified revised manuscript, including the abstract, to clearly discriminate our new results on anatomy and physiology of a novel OT pathway terminating in the vIPAG from the previously described OT circuit involving the spinal cord.

Also, the authors should revise the following questions.

1. The clinical studies showed that the analgesic effect of intranasal OT administration was different between genders. In Mai Iwasaki et al's study, adult female and male Wistar wild type and Sprague Dawley OTR-IRES-Cre rats (>8 weeks old; 250 - 350 g; Chronobiotron, Strasbourg, France) were used. It was wondered if the authors consider the sex difference in OT analgesia effect.

Indeed, clinical literature, as well as some animal research reports (e.g. elegant publications from Eagle Yi-Kung Huang lab), are rich in sex relevance in OT-induced analgesia. However, one might consider that clinical studies mostly rely on intra-nasal or intra-venous administration of exogenous OT, thus overwhelming any brain-area specific effect of OT, while most of sex differences pointed out in animals were revealed at spinal cord levels using, also, exogenous OT. In the present study focusing on a particular neuronal network, from PVN to vIPAG, we were not able to detect any difference between female and male when stimulating endogenous OT release from PVN axons in vIPAG nucleus. Indeed, while most of the behavioral tests were performed on female animals (**Figure 6**), we also assessed the efficacy of vIPAG OTR positive neurons to promote analgesia in males and found similar effects (**Supplementary Figure S10C**). We briefly discuss it in the discussion.

2. In “Materials and Methods, Stereotaxic injections”, the manuscript described “rAAVs were injected into the PVN, SON and vIPAG in different combinations, as needed by each experiment, and allowed to express for four weeks. The coordinates were chosen using the Paxinos rat brain atlas 41 (PVN: ML: +/-0.3 mm; AP: -1.4 mm; DV: -8.0 mm; SON: ML: +/-1.8 16mm; AP: -1.2 mm; DV: -9.25 mm; PAG: ML: +/-0.5 mm; AP: -7.0 mm; DV: -5.9/-5.0 mm)”. The methods were not easy, especially SON injection. The author should have provided anatomical proof.

We thank the reviewer for this important remark. We routinely inject PVN and SON with a high success rate and have published data and images of virus injections using the exact same coordinates before (Eliava et al., 2016, *Neuron*; Hasan et al., 2019, *Neuron*, Tang et al., 2020, *Nature Neuroscience* and Wahis et al., 2021, *Nature Neuroscience*). After each behavioral experiment, we carefully analyze all brains and confirm the correct injection sites. All animals that do not fulfill these rigorous criteria are excluded and not part of the final data set. We now provide images of SON and PVN injections of 6 WT rats (3 female and 3 male) to show the correct targeting of the respective nuclei using our mCherry/GFP-expressing viruses (please see Figure below). *The complete set of data can now be found on **Supplementary Figure S4**.*

3. Also, the manuscript described “Each site was injected with a total of 300 nL of viral solution via a glass pipette at a rate of 150 nl/min using a syringe pump”. I think that this injection process takes time. How to ensure the animal behaviors do not affect the brain nucleus injection ?

We apologize that we did not clearly communicate how our viral injections and subsequent behavioral experiments are performed. All stereotaxic virus injections are performed under ketamine/xylazine anesthesia, so no involuntary movements of the animals that could affect the precision of the injection occur during the surgery. After the virus injections, animals are given ample time for recovery and expression time of the viral proteins (3-4 weeks) so that there are no differences in baseline behavioral patterns between non-injected and injected animals. We now added a schematic depiction that shows the type of treatment, duration of virus expression and

allocation of rats to the different experimental conditions (**Supplementary Figure S3, please see below**).

Schematic depiction of experimental procedures

Reviewer #2:

The authors showed that a subset of oxytocin neurons project to the ventrolateral PAG and induce excitation of the majority of oxytocin receptor-expressing neurons in the ventrolateral PAG, leading to analgesia after inflammation or after chronic constriction of the sciatic nerve. Oxytocin has been shown to induce analgesia via acting on the dorsal root ganglion, spinal cord, PAG and insular cortex. This study further demonstrates that PAG oxytocin receptor-dependent analgesia is induced by activation of a new subset of oxytocin neurons. This is an interesting study, which will attract readers in a variety of fields. Following points should be considered.

1. Major finding of this study is demonstration of a novel subset of oxytocin neurons. Thus clear description of the distribution pattern of PVN oxytocin neurons projecting to the PAG as compared to neurons projecting to the spinal cord is helpful for readers. Is it possible to show a distribution map?

We thank the reviewer for this excellent suggestion and created a new supplementary figure highlighting the innervation of the PAG and spinal cord by parvocellular neurons that originate in the PVN. While we cannot make an absolute statement about the precise number of PAG-projecting PVN OT neurons due to limitations of the retrograde tracing technique, we attempted to gauge the approximate number of parvocellular OT cells that innervate PAG by comparing the number and intensity of OT fibers between PAG and spinal cord (see Eliava et al., *Neuron*, 2016). Given that the oxytocinergic innervation of the PAG appears to be higher by roughly 2-3-fold, we speculate that approximately 50-70 parvocellular OT cells might project to the vPAG compared to the approximately 30 parvocellular OT cells that we previously reported for the spinal cord. The respective schematic depiction is now shown as **Supplementary Figure S6, please see below.**

**Distribution map of distinct parvocellular oxytocinergic projection:
from PVN to PAG and SC**

2. This study demonstrate analgesic actions of PVN-PAG oxytocin system, In this study photic or chemogenic activation induced analgesia. Are there any evidences suggesting physiological

functions of the oxytocin system? The authors showed that painful stimuli activate oxytocin receptor-expressing neurons in the PAG. Why the oxytocin receptor antagonist had no effects in response to pain stimuli when no BL stimuli were applied?

We thank the reviewer for this interesting remark. Indeed, we made similar observation in most our studies, as colleagues from different labs. It is very interesting to note two specific points:

1) Stimulation of the endogenous OT system, as well as application of exogenous OT, has almost no effect in naïve animals, presenting no painful sensitization. This can be observed in various publications, including our own or those from Prof. Condes-Lara group, for example.

2) Antagonizing the OTR under painful sensitization rarely has an effect per se, as described in our past publications (Eliava et al., *Neuron*, 2016; Hilfiger et al., *Scientific Report*, 2021) and others (e.g. Prof. Condes-Lara or Eagle Yi-Kung Huang labs).

Therefore, the absence of acute effect of OTR antagonist might be due to the absence of spontaneous pain in most of the pain-induced sensitization models we use – the exact reason why ethical committees approve the use of such models. Thus, while acute painful stimulation induce OT release (occurring while pinching the paw of the animal for example, or shortly after CFA injection, during the acute phase of the inflammation), this might not occur spontaneously once the sensitization is established and the animal do not suffer spontaneous pain.

3. Details of painful stimuli in the experiments of Fos should be described.

Thanks for pointing this. We now include the detail of this experiment in Material and Methods section, as follow:

For the c-fos experiment, rats were pinched on the right hindpaw (with or without CFA) three times using the same forceps as for the mechanical pain threshold procedure. Progressive pressure was applied until the rat exhibited a pain reaction by either retracting its paw or squeaking. The control groups were handled in the same way but no pressure was applied. Rats were perfused 90 minutes later as described previously.

4. Majority of experiments were performed in female rats. Is there any reason?

Indeed, most of the experiments were performed in female rats. We did that by similitude toward our previously published studies (Eliava et al., *Neuron*, 2016; Hasan et al., *Neuron*, 2019, Tang et al., *Nature Neuroscience*, 2020). The two technical initial reasons for us to use female rats were i) the overall more active OT neurons in female, therefore leading to a more consistent expression of rAAV included proteins (Knobloch et al., *Neuron*, 2012) and ii) the stable size and weight of female Wistar rats by comparison to males, facilitating our numerous stereotaxic approaches and nociceptive sensitivities comparisons. Further anatomical analysis revealed limited, if any, difference of OT neurons projections between male and females (<https://www.biorxiv.org/content/10.1101/2022.01.17.476652v1.full>).

Moreover, one key experiment was reproduced in male Wistar rats, leading to similar results: we found that OTR expressing neurons activation in either male or female leads to a strong anti-hyperalgesic action (Figure 6 and S10).

5. Is there no influence of the phase of estrous cycle for behavioral pain response?

We thank the reviewer for raising this important, unexplored and misunderstood point. Very few studies were focused on the influence of estrous cycle in female per se. According to Anne Murphy's group, it seems to not influence the basal nociception of females, as depicted in the following figure from Loyd et al., J Neuroscience, 2008 showing similar thermal sensitivities during proestrus, estrus and diestrus.

However, estrus cycle seems to influence female sensitivity to exogenous modulation of hyperalgesia, as studied in the frame of opioid system. Indeed, Tyndalle group (Arguelles et al, Biochemical Pharmacology, 2022) showed that estrus cycle influence the level of available cytochrome P450 2D (CYP2D), as its concentration increased during estrus and decreased during diestrus. Thus, the oxycodone-induced analgesia (a mu opioid receptor agonist) is more less pronounced in estrus than in diestrus. While no dedicated study exist for the potential variation in oxytocin-induced analgesia during estrus cycle, we never observed such effect in the past (Eliava et al., 2016, *Neuron*).

6. A new transgenic rat line, OTR-IRES-Cre line was produced in this study. The rat line appears to be a knock-in line. However, a CRISPR/CAS method might have produced mis-integration. Was proper genomic insertion confirmed by performing DNA sequencing of PCR products amplified by a combination of a primer for upstream of the inserted site and a primer of downstream of the inserted site?

We thank the reviewer for this important remark. DNAs of Cre positive animals were further used for detection of homologous recombination at the OT locus. For this purpose the targeted region was amplified by PCR using the Q5 polymerase (NEB) with 100-200ng of genomic DNA as template. Primers were selected which bind up- and downstream of the insertion site (outside of the homology arms) and were each combined with a primer located within the IRES2-Cre construct. For the 5' insertion site the primers OT_upper_check (5' CTCCCAGGGAAGATCTGTACC 3') together with IRES2_rev (5' GCTTCGGCCAGTAACGTTAGG 3') and for the 3' site the primers CRISPR_bGH_for

OXTR_upper check_for:
CTCCCAGGGAAGATCTGTACC

CRISPR_BGH_FOR:
gacaatagcaggcatgctgg

+

+

IRES2_rev:
gCTTCGGCCAGTAACGTTAGG
= 1394 bp

OXTR_lower check_rev
GGGTTTTCCAGAACTCAGC
= 1640 bp

(5' gacaatagcagggcatgctgg3') together with OT_rev_hom (5' GGGTTTTCCAGAACTCAGC 3') were used. This is now included in **Supplementary Figure S1**.

7. The number of rats used for detection of Cre mRNA and OTR mRNA should be described clearly.

We thank the reviewer for this important comment and apologize for not providing this information initially. We analyzed brain sections from n=3 rats (3 sections each) and now provide additional images and quantification on **Supplementary Figure S2A-B**.

Figure S2

A

B

C

D

E

F

To validate that our viruses are reliably expressed in a Cre-dependent manner using our novel OTR-Cre line, we injected rats (n=3) with a cre-dependent AAV expressing GFP and found that virtually all (>96%) virus-expressing cells were also Cre-positive (**Supplementary Figure S2C-D**). To further confirm the specificity of our novel transgenic OTR-Cre rat model, we performed a Western blot against the Cre protein (39kDa) in wildtype and knock-in animals and detected the Cre protein band only in knock-in rats. The respective western blot was repeated in five separate reactions, with the same result (**Supplementary Figure S2E-F**).

8. All PVN-PAG oxytocin projections are ipsilateral? Information of laterality should be described clearly. Is there any difference in anatomy or physiology between left and right sides? How many rats received left side injection or right side injection for anatomical or behavioral experiments?

We thank the reviewer for this important comment and injected 6 rats (n=3 males and n=3 females) unilaterally into the PVN. We then analyzed the PAG of injected rats at various bregma levels (-6.5mm, -7.5mm and -8.5mm), but - to our surprise - did not find any indication for oxytocin fiber laterality, as both the ipsi and contralateral side of the vPAG were densely innervated by

fibers originating from the unilateral PVN injection. Thus, we conclude that each PVN innervates the vPAG in both the right and left hemispheres. The image below shows the unilateral injections of virus into SON and PVN, expression of the viruses in the respective nuclei and labeling of OT fibers in right and left vPAG. The complete set of data can now be found on **Supplementary Figure S4**.

9. Activity of a small number of PAG OTR-expressing neurons was suppressed by stimulation of oxytocin neurons. Both excitatory and inhibitory actions were blocked by an oxytocin receptor antagonist? Possible explanation of no response or inhibitory response is helpful for readers.

We apologize for the apparent lack of clarity in our results display. Indeed, we found oT-induced inhibition of vPAG neurons in 2 neurons (**Figure 4**). However, in the course of those *in vivo* electrophysiological recordings we did not assess the nature of vPAG neurons recorded. However, we found that most of vPAG OTR-expressing neurons seems to be GABAergic (Figure 1). Given that our *ex vivo* recordings indicate that virtually all OTR expressing neurons are activated upon OT binding (Figure 1), the 2 recorded inhibition might be the n+1 neurons inhibited by OTR-GABA interneurons, putatively projection neurons to downstream pathways through the spinal cord. We now discuss this in the manuscript

10. The expression “evoked OT release in vPAG” was used in the paper. However, OT release was not directly examined. Stimulation of OT fibers does not necessarily mean OT release. Precise expression should be used.

The reviewer is right. Therefore, we now performed direct examination of OT release using the newly developed fluorescent G-protein-coupled receptor-activation-based (GRAB) for OT by Yulong Li lab, which was also tested and verified by us (Qian et al., <https://doi.org/10.1101/2022.02.10.480016>). We now show that optogenetic activation of OT neurons leads to actual OT release in the vPAG, as described in the new **Figure 4A-F** as thereafter.

11. Is there any evidence that blue light application to the PAG did not activate passing fibers projecting to the spinal cord?

We thank the reviewer for this important question. In fact, OTergetic projections arising in the PVN that innervate the PAG and spinal cord are two distinct neuronal populations (**Figure S5 and S6**).

Moreover, neuroanatomical studies from our lab using whole brain clearing methods (Eliava et al., 2016, *Neuron*, Figure S2) and other conventional studies using CTB retrograde tracing and DAB staining (Geerling et al., 2010, *J Comp Neurol*, Figure 4, PMID: 20187136) clearly show that OTergetic axons traverse the ventral surface of the brain to innervate the spinal cord, but OTergetic axons innervating the vPAG branch off as a separate fiber bundle around Bregma -5.5mm.

In addition, we performed double immunohistochemistry against OT and synaptophysin (SYN+) to quantify the number of SYN+ OTergic fibers within the vPAG. We found that that 80% of OTergic fibers at Bregma -6.5mm, 90% of OTergic fibers at Bregma -7.5mm and almost 100% of OTergic fibers at Bregma -8.5mm were positive for SYN+, thus essentially ruling out that these fibers further project to the spinal cord and corroborating the previous, above-mentioned neuroanatomical studies. We now present the new results on **Supplemental Figure S7**, please see the newly added graph below.

12. In the experiments with an oxytocin receptor antagonist injection into the PAG, vehicle was injected for the control group (Fig 5 C1)? If so, in figure 5 C1 “ChR2 BL” should read “ChR2 BL + Vehicle”.

13. “we found that vPAG-BL had no effect on mechanical sensitivity when testing the contralateral paw (Figure 6D2)” ”mechanical stimuli 195.7... to 265.3 (Figure S6A1-2)..... Figure S6A2: n=7.” Figure numbers are correct?

We thank the reviewer for these observations. As we included new figures we now change figure appealing and hope to have not mis-target them in the current version of the manuscript.

Reviewer #3:

In this manuscript, Iwasaki et al. investigate the role of oxytocin (OCT) expressing cells in the paraventricular (PVN) nuclei in the descending analgesic pathway. For this, they generated rats for which Cre-expression was driven by production of the oxytocin receptor (OCT-Cre rats). Using ex-vivo slice electrophysiology, they confirm that OCT-expressing neurons in PAG (labeled with GFP via AAV-DIO injection) increase firing in response to the selective OTR agonist TGOT. They used anterograde and retrograde techniques to show that OT neurons in PVT consist of distinct populations sending projections to the ventrolateral periaqueductal gray (vlPAG) and the spinal cord (SC). Optogenetic activation of PVN-OT axons in vivo within the vlPAG led to increased activity of approximately 26% of neurons recorded within the vlPAG. More in depth in-vivo studies reveal optogenetic activation of PVN-OT axons within the vlPAG could reduce wide dynamic range (WDR) neuronal responses evoked by electrical stimulation of the hind paw. Lastly, Iwasaki et al. shows that this same stimulation of PVN-OT axons in the vlPAG can reduce sensory pain behavior in models of inflammatory and neuropathic pain; an effect which they report is sensitive to the OTR antagonist L-368,899. Overall, this exciting and novel study provides new insight into a unique OT pathway that can modulate noxious input via the vlPAG. This is important for the development of novel analgesics that could target OT circuits. However, there are a number of concerns that limit enthusiasm, which are discussed below:

1. The representative ephys trace in Figure S1E1 clearly shows that TGOT increases firing frequency in response to a depolarizing step current, yet this is not mentioned. The authors just focus on latency to first spike, which may be biologically difficult to alter given that baseline latency is relatively rapid.

We thank the reviewer for this observation. We now include this simple analysis in **Figure S1**, and show that TGOT has in average no effect of current step induced spiking activity of OTR- neurons. The new graph is presented thereafter. However, the number of recorded cells seems too small to perform more extensive analysis in this frame and deliver a strong conclusion – further experiments exploring the early change in neuronal electrical properties following OTR activation will be needed.

2. It is unclear why the authors did not use ex-vivo slice electrophysiology to test the effects of ChR2+ OT+ axons on recorded neurons in the vlPAG. This would allow for the identification of OTR+ neurons using TGOT, and the use of OTR antagonists or glutamate antagonists to confirm that any changes to excitability were due to release of OT from ChR2+ axons. This would also confirm that OT can be released via ChR2+ stimulation as the effects seen in Figure 4 could be due to release of glutamate (not OT) from the small population of VGluT2+ OT axons. Figure 5 addresses some of this, but using dOVT in this paradigm would confirm the cellular mechanism.

The reviewer is right, the suggested experiments would be ideal and correspond to what we previously published (Knobloch et al., Neuron, 2012; Hasan et al., Neuron, 2019). Following reviewer suggestion, we attempted to performed ex vivo recordings of vlPAG OTR expressing neurons and assessed their response to ChR2-induced activation of OT neurons. Unfortunately, for unforeseen reason we were not able to established a stable enough patch clamp recording to assess effect of ChR2 stimulation followed by pharmacological assessment of the components of this response.

However, we believe our results presented in **Figure 5**, showing that vIPAG dOVT infusion prevented the effect of ChR2-induced activation of OT axons within the vIPAG in spinal cord neurons activity is a good indication of the OT component of the effect.

Moreover, to address the point of actual OT release triggered by OT axons stimulation, we now performed direct examination of OT release using the newly developed fluorescent G-protein-coupled receptor-activation-based (GRAB) for OT by Yulong Li lab (Qian et al., <https://doi.org/10.1101/2022.02.10.480016>). This allowed us to reveal that optogenetic activation of OT neurons leads to actual OT release in the vIPAG, as described in the new **Figure 4** as thereafter.

3. In regards to the previous comment, it is a bit of a stretch to conclude the BL-evoked OT release leads to an overall excitation of putative OTR+ vIPAG neurons.

Please see our new experiment showing that ChR2 activation indeed lead to OT release, as measured with GRAB_{OTR} sensor.

4. The explanation of Figure 6D2 in the text does not match what is presented in the Figure. The figure shows what appears to be an experiment using the chronic constriction injury, while the text is referring to absence of vIPAG-BL effect in the paw contralateral to CFA injection. This is the same for Figure S6A1-A2.

We thank the reviewer for these observations. As we included new figures, we now change figure call and hope to have not mis-target them in the current version of the manuscript.

5. The intermittent use of male and female rats is not well justified. While it seems a majority of rats used were female, it is unclear why males were introduced only in some of the experiments. Furthermore, sex-specific differences were not explored nor mentioned, which is of interest considering reported sex differences in clinical OT studies.

The reviewer is right we didn't paid enough attention to this point in our initial manuscript; we now explain more clearly what sex was used.

In our study, most of the experiments were performed in female rats. We did that by similitude toward our previously published studies (Eliava et al., Neuron, 2016; Hasan et al., Neuron, 2019,

Tang et al., Nature Neuroscience, 2020). The two technical initial reasons for us to use female rats were i) the overall more active OT neurons in female, therefore leading to a more consistent expression of rAAV included proteins (Knobloch et al., Neuron, 2012) and ii) the stable size and weight of female Wistar rats by comparison to males, facilitating our numerous stereotaxic approaches and nociceptive sensitivities comparisons. Further anatomical analysis revealed limited, if any, difference of OT neurons projections between male and females (<https://www.biorxiv.org/content/10.1101/2022.01.17.476652v1.full>).

Indeed, clinical literature, as well as some animal research reports (e.g. elegant publications from Eagle Yi-Kung Huang lab), are rich in sex relevance in OT-induced analgesia. However, one might consider that clinical studies mostly rely on intra-nasal or intra-venous administration of exogenous OT, thus overwhelming any brain-area specific effect of OT, while most of sex differences pointed out in animals were revealed at spinal cord levels using, also, exogenous OT. In the present study focusing on a particular neuronal network, from PVN to vIPAG, we were not able to detect any difference between female and male when stimulating endogenous OT release from PVN axons in vIPAG nucleus. Indeed, while most of the behavioral tests were performed on female animals (**Figure 6**), we also assessed the efficacy of vIPAG OTR positive neurons to promote analgesia in males and found similar effects (**Supplementary Figure S10C**). We briefly discuss it in the discussion.

6. Statistical analyses is rather vaguely defined. Mention of tests used for analysis in Figure legends including t and F values would be more informative.

All statistical analysis were reviewed and better reported in the text. We thank the reviewer for pointing that out.

REVIEWER COMMENTS

Reviewer #2 (Remarks to the Author):

The authors revised the manuscript adequately. No further comments.

Reviewer #3 (Remarks to the Author):

While the reviewers addressed some of my comments, there are still some issues that remain concerning.

1. The authors state “Unfortunately, for unforeseen reasons we were not able to establish a stable enough patch clamp recording to assess the effect of ChR2 stimulation followed by pharmacological assessment of the components of this response.” This is a rather vague response. The authors are capable of doing these experiments as they showed with intrinsic recordings, so it is unclear why stable recordings were not doable to address my previous concern.
2. While the GRAB experiments are very exciting, little detail is given regarding rigor (i.e. negative controls) to validate that the sensor is actually measuring OT levels.
3. The Supplementary Table is disorganized. It looks as if it were just cut and paste from statistics software, and is difficult for a reader to follow.

Answers to Reviewers' Comments

We again thank the reviewers for their time and effort in evaluating our manuscript. We regret the absence of Reviewer #1 feedback, as her/his initial remarks were very interesting. For the reviewers' convenience, our responses here as well as the new data in the revised manuscript are typed in blue.

Reviewer #2

The authors revised the manuscript adequately. No further comments.

We deeply thank the reviewer for her/his positive evaluation.

Reviewer #3 (Remarks to the Author):

While the reviewers addressed some of my comments, there are still some issues that remain concerning.

1. The authors state "Unfortunately, for unforeseen reason we were not able to establish a stable enough patch clamp recording to assess effect of ChR2 stimulation followed by pharmacological assessment of the components of this response." This is a rather vague response. The authors are capable of doing these experiments as they showed with intrinsic recordings, so it is unclear why stable recordings were not doable to address my previous concern.
2. It is unclear why the authors did not use ex-vivo slice electrophysiology to test the effects of ChR2+ OT+ axons on recorded neurons in the vIPAG. This would allow for the identification of OTR+ neurons using TGOT, and the use of OTR antagonists or glutamate antagonists to confirm that any changes to excitability were due to release of OT from ChR2+ axons. This would also confirm that OT can be released via ChR2+ stimulation as the effects seen in Figure 4 could be due to release of glutamate (not OT) from the small population of VGluT2+ OT axons. Figure 5 addresses some of this, but using dOVT in this paradigm would confirm the cellular mechanism.

The reviewer is right, our lab is equipped to perform such recordings, as we did in the past in other brain regions (e.g. the central amygdala in Hasan et al., *Neuron*, 2019 and Wahis et al., *Nature Neuroscience* 2021). *Ex vivo* patch-clamp recordings were used to record the response of GFP positive neurons (identified as expressing OTR) to bath application of TGOT. However, the subsequent optogenetic experiments on PAG OTR+ neurons have failed.

Indeed, following our pharmacological results, we attempted to perform the exact experiment described by the reviewer: we injected a rAAV OTp-ChR2 in the PVN and a rAAV DIO-mCherry in the vIPAG of OTR-IRES-Cre rats, allowing us to stimulate vIPAG OTaxons and record identified OTR neurons. Under these conditions, and despite a second set of experiment for this round of revision, we were not able to obtain a stable patch clamp for the >30min of recordings requested

on a regular basis. Although in very rare cases, we were able to have stable recording and observe a putative response to OT axons stimulation, these events occurred too rarely and were not powerful enough to perform further pharmacological experiments aiming at deciphering the respective contribution of OT and glutamate release in the vIPAG. For this reason, we decided to not include these data: we believe that it might only be misleading for future readers to see a partial experiment involving a very few neurons. We hope the reviewer would agree with this point.

Among the potential reasons for this technical failure, we want to mention:

- The difficulty to perform a stable patch-clamp recording in vIPAG of adult rats on identified neurons, as despite all our expertise, fluorescent neurons are located deep in the slice and remain very fragile: it is difficult to obtain a stable patch and patched neurons often die within minutes.
- The age of the animals is a second intrinsic aspect of the preparation that we cannot overlook in the current situation; indeed, the adult animals used for this study are at least 7 weeks old, while most of the stable patch-clamp recordings are usually obtained in younger animals of ~3 weeks old.
- We might face a technical limitation of slice recording: we counted only very few axons innervating vIPAG (Figure 3B). While this limited number of axons lead to a significant release of oxytocin in slice (Figure 4A-F) and a significant increase in firing of ~25% of vIPAG neurons recorded in vivo (Figure 4G-K), there is a possibility that the slicing process impairs the integrity of numerous OT axons. This might explain why we only rarely recorded a response in identified OTR neurons upon OT axon stimulation in patch-clamp when successfully patched. On the other hand, this illustrates the need of complementary approaches when tackling a biological phenomenon, as exemplified with *ex vivo* and *in vivo* strategies.

In conclusion, we recognize our failure to perform optogenetic driven deciphering of OT / glutamate in the vIPAG region. In our hands, it is the second time such a failure occurs, the first one being the description of the OT PVN → SON connections described in Eliava et al., Neuron, 2016. In this study, we also failed to record in *ex vivo* patch-clamp stable response to OT axons stimulations, while in vivo recordings generated convincing results and anatomical analysis clearly showed close axo-somatic contacts. Fully tackling the OT / Glutamate mechanism represents an endeavor that would require a 2 years independent post-doc project. In the course of such study, we should have to first focus on a brain region that is better explored, such as the central amygdala, as we partially did in Hasan et al., Neuron, 2019. We hope the reviewer will understand these limitations. For transparency reason, we added a sentence in the discussion mentioning that the general OT / glutamate co-release mechanism remains to be elucidated, as thereafter:

'Of note, we did not decipher the functional involvement of a putative OT / glutamate co-release in this region, a mechanism of general interest for cellular network modulation that remains to be elucidated.'

Despite the lack of sustained *ex vivo* patch recording data, we believe that our results presented in Figure 4A-F, showing the actual release of OT upon stimulation of OT axons, are supported by the ones presented in Figure 5C-D, demonstrating that vIPAG dOVT infusion prevented the effect of ChR2-induced activation of OT axons within the vIPAG in spinal cord neurons activity. This is

good indication of the at least existence of OT component of the effect. Therefore, we are convinced that the scientific conclusions of our present manuscript, describing in details a new OT PVN → vIPAG → Spinal Cord pathway controlling nociception, are not affected by the absence of the requested ex vivo patch clamp results.

2. While the GRAB experiments are very exciting, little detail is given regarding rigor (i.e. negative controls) to validate that sensor is actually measuring OT levels.

We fully agree with the reviewer. Following her/his recommendation, we performed two different controls in the scope of this revision. First and as a negative control, we performed the same optic stimulation and fluorescent recording in absence of C1V1 expressed in OT axons. The results displayed in gray in **Figure 4D-F** show a slight increase of signal after stimulation. Second, we performed OT axons stimulation in presence of atosiban, used here as an antagonist, as previously characterized by Prof. Li lab in mice with our subsequent analysis of the OT sensor in rats (Qian et al., Nature Biotechnology, in press), which are the subjects of the present manuscript. As the reviewer may see, the results shown in red in **Figure 4D-F** almost exactly follow the negative control curve. Altogether, this experimental approach now convincingly shows that optogenetic activation of OT axons in the vIPAG triggers actual release of OT. We hope that the reviewer agrees with our conclusion.

3. The Supplementary Table is disorganized. It looks as if it were just cut and paste from statistics software, and is difficult for a reader to follow.

We thank the reviewer for pointing this. We now worked on Supplementary Table reorganization and hope it is now more understandable for the reader. In particular, we reviewed the display of Figure tables, separating “data sets” and “statistical analysis”.

REVIEWER COMMENTS

Reviewer #3 (Remarks to the Author):

The authors addressed my experimental concerns.

However, the Supplemental Table remains difficult to follow. It would be beneficial to present the table in a Landscape orientation.

Answers to Reviewers' Comments

We again thank the reviewers for their time and effort in evaluating our manuscript.

Reviewer #3 (Remarks to the Author):

The authors addressed my experimental concerns.

However, the Supplemental Table remains difficult to follow. It would be beneficial to present the table in a Landscape orientation.

It seems that the formatting issue brought up by the reviewer appears to stem from a suboptimal Excel-> PDF conversion. After discussion with the editor, we agreed to maintain the file as its current form.